# NodeFormer: A Scalable Graph Structure Learning Transformer for Node Classification

**Qitian Wu**[1], **Wentao Zhao**[1], **Zenan Li**[1], **David Wipf**[2], **Junchi Yan**[1]*

[1]Department of Computer Science and Engineering, Shanghai Jiao Tong University
[2]Amazon Web Service, Shanghai AI Lab
`{echo740,permanent,emiyali,yanjunchi}@sjtu.edu.cn, davidwipf@gmail.com`

## Abstract

Graph neural networks have been extensively studied for learning with inter-connected data. Despite this, recent evidence has revealed GNNs' deficiencies related to over-squashing, heterophily, handling long-range dependencies, edge incompleteness and particularly, the absence of graphs altogether. While a plausible solution is to learn new adaptive topology for message passing, issues concerning quadratic complexity hinder simultaneous guarantees for scalability and precision in large networks. In this paper, we introduce a novel all-pair message passing scheme for efficiently propagating node signals between arbitrary nodes, as an important building block for a pioneering Transformer-style network for node classification on large graphs, dubbed as NODEFORMER. Specifically, the efficient computation is enabled by a kernerlized Gumbel-Softmax operator that reduces the algorithmic complexity to linearity w.r.t. node numbers for learning latent graph structures from large, potentially fully-connected graphs in a differentiable manner. We also provide accompanying theory as justification for our design. Extensive experiments demonstrate the promising efficacy of the method in various tasks including node classification on graphs (with up to 2M nodes) and graph-enhanced applications (e.g., image classification) where input graphs are missing. The codes are available at `https://github.com/qitianwu/NodeFormer`.

## 1 Introduction

Relational structure inter-connecting instance nodes as a graph is ubiquitous from social domains (e.g., citation networks) to natural science (protein-protein interaction), where graph neural networks (GNNs) [32, 19, 14, 36] have shown promising power for leveraging such data dependence as geometric priors. However, there arises increasing evidence challenging the core GNN hypothesis that propagating information along observed graph structures will necessarily produce better node-level representations for prediction on each individual instance node. Conflicts with this premise lead to commonly identified deficiencies with GNN message-passing rules w.r.t. heterophily [53], over-squashing [2], long-range dependencies [8], and graph incompleteness [11], etc.

Moreover, in graph-enhanced applications, e.g., text classification [46], vision navigation [12], physics simulation [30], etc., graph structures are often unavailable though individual instances are strongly inter-correlated. A common practice is to artificially construct a graph via some predefined rules (e.g., $k$-NN), which is agnostic to downstream tasks and may presumably cause the misspecification of GNNs' inductive bias on input geometry (induced by the local feature propagation design).

Natural solutions resort to organically combining learning optimal graph topology with message passing. However, one critical difficulty is the *scalability* issue with $O(N^2)$ (where $N$ denotes

---

*Wentao Zhao and Zenan Li contribute equally. The SJTU authors are also with MoE Key Lab of Artificial Intelligence, SJTU. Junchi Yan is the correspondence author who is also with Shanghai AI Laboratory.

36th Conference on Neural Information Processing Systems (NeurIPS 2022).

#nodes) computational complexity, which is prohibitive for large networks (with $10K \sim 1M$ nodes). Some existing approaches harness neighbor sampling [51], anchor-based adjacency surrogates [4] and hashing schemes [43] to reduce the overhead; however, these strategies may sacrifice model precision and still struggle to handle graphs with million-level nodes. Another obstacle lies in the increased degrees of freedom due to at least an $N \times N$ all-pair similarity matrix, which may result in large combinatorial search space and vulnerability to over-fitting.

In this work, we introduce a novel all-pair message passing scheme that can scale to large systems without compromising performance. We develop a kernelized Gumbel-Softmax operator that seamlessly synthesizes *random feature map* [27] and approximated sampling strategy [16], for distilling latent structures among all the instance nodes and yielding moderate gradients through differentiable optimization. Though such a combination of two operations involving randomness could potentially result in mutual distortion, we theoretically prove that the new operator can still guarantee a well-posed approximation for concrete variables (discrete structures) with the error bounded by feature dimensions. Furthermore, such a design can reduce the algorithmic complexity of learning new topology per layer to $O(N)$ by avoiding explicit computation for the cumbersome all-pair similarity.

The proposed module opens the door to a new class of graph networks, i.e., NODEFORMER (*Scalable Transformers for Node Classification*), that is capable of efficiently propagating messages between arbitrary node pairs in flexible layer-specific latent graphs. And to accommodate input graphs (if any), we devise two simple techniques: a relational bias and an edge-level regularization loss, as guidance for properly learning adaptive structures. We evaluate our approach on diverse node classification tasks ranging from citation networks to images/texts. The results show its promising power for tackling heterophily, long-range dependencies, large-scale graphs, graph incompleteness and the absence of input graphs. The contributions of this paper are summarized as follows:

• We develop a kernelized Gumbel-Softmax operator which is proven to serve as a well-posed approximation for concrete variables, particularly the discrete latent structure among data points. The new module can reduce the algorithmic complexity for learning new message-passing topology from quadratic to linear w.r.t. node numbers, without sacrificing the precision. This serves as a pioneering model that successfully scales graph structure learning to large graphs with million-level nodes.

• We further propose NODEFORMER, a new class of graph networks with layer-wise message passing as operated over latent graphs potentially connecting all nodes. The latter are optimized in an end-to-end differentiable fashion through a new objective that essentially pursues sampling optimal topology from a posterior conditioned on node features and labels. To our knowledge, NODEFORMER is the first Transformer model that scales all-pair message passing to large node classification graphs.

• We demonstrate the model's efficacy by extensive experiments over a diverse set of datasets, including node classification benchmarks and image/text classification, where significant improvement over strong GNN models and SOTA structure learning methods is shown. Besides, it successfully scales to large graph datasets with up to 2M nodes where prior arts failed, and reduces the time/space consumption of the competitors by up to 93.1%/80.6% on moderate sized datasets.

## 2 Related Works

**Graph Neural Networks.** Building expressive GNNs is a fundamental problem in learning over graph data. With Graph Attention Networks (GAT) [36] as an early attempt, there are many follow-up works, e.g., [22, 42], considering weighting the edges in input graph for enhancing the expressiveness. Other studies, e.g., [28, 52] focus on sparsifying input structures to promote robust representations. There are also quite a few approaches that propose scalable GNNs through, e.g., subgraph sampling [48], linear feature mapping [39], and channel-wise transformation [49], etc. However, these works cannot learn new edges out of the scope of input geometry, which may limit the model's receptive fields within local neighbors and neglect global information.

**Graph Structure Learning.** Going beyond observed topology, graph structure learning targets learning a new graph for message passing among all the instances [54]. One line of work is similarity-driven where the confidence of edges are reflected by some similarity functions between node pairs, e.g., Gaussian kernels [43], cosine similarity [4], attention networks [17], non-linear MLP [7] etc. Another line of work optimizes the adjacency matrix. Due to the increased optimization difficulties, some sophisticated training methods are introduced, such as bi-level optimization [11], variational

Table 1: Comparison of popular graph structure learning approaches for *node-level tasks* where in particular, the graph connects all instance nodes and one's target is for prediction on each individual node. For *parameterization*, 'Function' means learning through functional mapping and 'Adjacency' means directly optimizing graph adjacency. For *expressivty*, 'Fixed' means learning one graph shared by all propagation layers and 'Layer-wise' means learning graph structures per layers. The *largest demo* means the largest # nodes of datasets used. † $m$ denotes # anchors (i.e., a subset of nodes).

| Models | Parameterization | Expressivity | Input Graphs | Inductive | Complexity | Largest Demo |
|---|---|---|---|---|---|---|
| LDS-GNN [11] | Adjacency | Fixed | Required | No | $O(N^2)$ | 0.01M |
| ProGNN [18] | Adjacency | Fixed | Required | No | $O(N^2)$ | 0.02M |
| VGCN [10] | Adjacency | Fixed | Required | No | $O(N^2)$ | 0.02M |
| BGCN [51] | Adjacency | Fixed | Required | No | $O(N^2)$ | 0.02M |
| GLCN [17] | Function | Fixed | Not necessary | Yes | $O(N^2)$ | 0.02M |
| IDGL [4] | Function | Fixed | Required | Yes | $O(N^2)$ or $O(Nm)^\dagger$ | 0.1M |
| NODEFORMER (Ours) | Function | Layer-wise | Not necessary | Yes | $O(N)$ or $O(E)$ | 2M |

approaches [10, 20], Bayesian inference [51] and projected gradient descent [18]. To push further the limits of structure learning, this paper proposes a new model NODEFORMER (for enabling scalable node-level Transformers) whose merits are highlighted via a high-level comparison in Table 1. In particular, NODEFORMER enables efficient structure learning in each layer, does not require input graphs and successfully scales to graphs with 2M nodes.

**Node-Level v.s. Graph-Level Prediction.** We emphasize upfront that our focus is on *node-level* prediction tasks involving a single large graph such that scalability is paramount, especially if we are to consider arbitrary relationships across *all* nodes (each node is an instance with label and one can treat all the nodes non-i.i.d. generated due to the inter-dependence) for structure-learning purposes. Critically though, this scenario is quite distinct from *graph-level* classification tasks whereby each i.i.d. instance is itself a small graph and fully connecting nodes *within* each graph is computationally inexpensive. While this latter scenario has been explored in the context of graph structure learning [38] and all-pair message passing design, e.g., graph Transformers [9], existing efforts do not scale to the large graphs endemic to node-level prediction.

## 3   NODEFORMER: A Transformer Graph Network at Scale

Let $\mathcal{G} = (\mathcal{N}, \mathcal{E})$ denote a graph with $\mathcal{N}$ a node set ($|\mathcal{N}| = N$) and $\mathcal{E} \subseteq \mathcal{N} \times \mathcal{N}$ an edge set ($|\mathcal{E}| = E$). Each node $u \in \mathcal{N}$ is assigned with node features $\mathbf{x}_u \in \mathbb{R}^D$ and a label $y_u$. We define an adjacency matrix $\mathbf{A} = \{a_{uv}\} \in \{0,1\}^{N \times N}$ where $a_{uv} = 1$ if edge $(u,v) \in \mathcal{E}$ and $a_{uv} = 0$ otherwise. Without loss of generality, $\mathcal{E}$ could be an empty set in case of no input structure. There are two common settings: transductive learning, where testing nodes are within the graph used for training, and inductive learning which handles new unseen nodes out of the training graph. The target is to learn a function for node-level prediction, i.e., estimate labels for unlabeled or new nodes in the graph.

**General Model and Key Challenges.** We start with the observation that the input structures may not be the ideal one for propagating signals among nodes and instead there exist certain latent structures that could facilitate learning better node representations. We thus consider the updating rule

$$\tilde{\mathbf{A}}^{(l)} = g(\mathbf{A}, \mathbf{Z}^{(l)}; \omega), \quad \mathbf{Z}^{(l+1)} = h(\tilde{\mathbf{A}}^{(l)}, \mathbf{A}, \mathbf{Z}^{(l)}; \theta), \tag{1}$$

where $\mathbf{Z}^{(l)} = \{\mathbf{z}_u^{(l)}\}_{u \in \mathcal{N}}$ and $\tilde{\mathbf{A}}^{(l)} = \{\tilde{a}_{uv}^{(l)}\}_{u,v \in \mathcal{N}}$ denotes the node representations and the estimated latent graph of the $l$-th layer, respectively, and $g, h$ are both differentiable functions aiming at 1) structure estimation for a layer-specific latent graph $\tilde{\mathbf{A}}^{(l)}$ based on node representations and 2) feature propagation for updating node representations, respectively. The model defined by Eqn. 1 follows the spirit of Transformers [35] (where in particular $\tilde{\mathbf{A}}^{(l)}$ can be seen as an attentive graph) that potentially enables message passing between any node pair in each layer, which, however, poses two *challenges*:

• **(Scalability)**: How to reduce the prohibitive quadratic complexity for learning new graphs?

• **(Differentiability)**: How to enable end-to-end differentiable optimization for discrete structures?

Notice that the first challenge is non-trivial in node-level prediction tasks (the focus of our paper), since the latent graphs could potentially connect *all the instance nodes* (e.g., from thousands to millions, depending on dataset sizes), which is fairly hard to guarantee both precision and scalability.

## 3.1 Efficient Learning Discrete Structures

We describe our new message-passing scheme with an efficient kernelized Gumbel-Softmax operator to resolve the aforementioned challenges. We assume $\mathbf{z}_u^{(0)} = \mathbf{x}_u$ as the initial node representation.

**Kernelized Message Passing.** We define a full-graph attentive network that estimates latent interactions among instance nodes and enables corresponding densely-connected message passing:

$$\tilde{a}_{uv}^{(l)} = \frac{\exp((W_Q^{(l)}\mathbf{z}_u^{(l)})^\top(W_K^{(l)}\mathbf{z}_v^{(l)}))}{\sum_{w=1}^N \exp((W_Q^{(l)}\mathbf{z}_u^{(l)})^\top(W_K^{(l)}\mathbf{z}_w^{(l)}))}, \quad \mathbf{z}_u^{(l+1)} = \sum_{v=1}^N \tilde{a}_{uv}^{(l)} \cdot (W_V^{(l)}\mathbf{z}_v^{(l)}), \quad (2)$$

where $W_Q^{(l)}$, $W_K^{(l)}$ and $W_V^{(l)}$ are learnable parameters in $l$-th layer. We omit non-linearity activation (after aggregation) for brevity. The updating for $N$ nodes in one layer using Eqn. 2 requires prohibitive $\mathcal{O}(N^2)$ complexity. Also, given large $N$, the normalization in the denominator would shrink attention weights to zero and lead to gradient vanishing. We call this problem as *over-normalizing*.

To accelerate the full-graph model, we observe that the *dot-then-exponentiate* operation in Eqn. 2 can be converted into a pairwise similarity function:

$$\mathbf{z}_u^{(l+1)} = \sum_{v=1}^N \frac{\kappa(W_Q^{(l)}\mathbf{z}_u^{(l)}, W_K^{(l)}\mathbf{z}_v^{(l)})}{\sum_{w=1}^N \kappa(W_Q^{(l)}\mathbf{z}_u^{(l)}, W_K^{(l)}\mathbf{z}_w^{(l)})} \cdot (W_V^{(l)}\mathbf{z}_v^{(l)}), \quad (3)$$

where $\kappa(\cdot, \cdot) : \mathbb{R}^d \times \mathbb{R}^d \to \mathbb{R}$ is a positive-definite kernel measuring the pairwise similarity. The kernel function can be further approximated by random features (RF) [27] which serves as an unbiased estimation via $\kappa(\mathbf{a}, \mathbf{b}) = \langle \Phi(\mathbf{a}), \Phi(\mathbf{b}) \rangle_\mathcal{V} \approx \phi(\mathbf{a})^\top \phi(\mathbf{b})$, where the first equation is by Mercer's theorem with $\Phi : \mathbb{R}^d \to \mathcal{V}$ a basis function and $\mathcal{V}$ a high-dimensional vector space, and $\phi(\cdot) : \mathbb{R}^d \to \mathbb{R}^m$ is a low-dimensional feature map with random transformation. There are many potential choices for $\phi$, e.g., Positive Random Features (PRF) [6]

$$\phi(\mathbf{x}) = \frac{\exp\left(\frac{-\|\mathbf{x}\|_2^2}{2}\right)}{\sqrt{m}}[\exp(\mathbf{w}_1^\top\mathbf{x}), \cdots, \exp(\mathbf{w}_m^\top\mathbf{x})], \quad (4)$$

where $\mathbf{w}_k \sim \mathcal{N}(0, I_d)$ is i.i.d. sampled random transformation. The RF converts dot-then-exponentiate operation into inner-product in vector space, which enables us to re-write Eqn. 3 (assuming $\mathbf{q}_u = W_Q^{(l)}\mathbf{z}_u^{(l)}$, $\mathbf{k}_u = W_K^{(l)}\mathbf{z}_u^{(l)}$ and $\mathbf{v}_u = W_V^{(l)}\mathbf{z}_u^{(l)}$ for simplicity):

$$\mathbf{z}_u^{(l+1)} = \sum_{v=1}^N \frac{\phi(\mathbf{q}_u)^\top\phi(\mathbf{k}_v)}{\sum_{w=1}^N \phi(\mathbf{q}_u)^\top\phi(\mathbf{k}_w)} \cdot \mathbf{v}_v = \frac{\phi(\mathbf{q}_u)^\top\sum_{v=1}^N \phi(\mathbf{k}_v) \cdot \mathbf{v}_v^\top}{\phi(\mathbf{q}_u)^\top\sum_{w=1}^N \phi(\mathbf{k}_w)}. \quad (5)$$

The key advantage of Eqn. 5 is that the two summations are shared by each $u$, so that one only needs to compute them once and re-used for others. Such a property enables $\mathcal{O}(N)$ computational complexity for full-graph message passing, which paves the way for learning graph structures among large-scale instances. Moreover, one can notice that Eqn. 5 avoids computing the $N \times N$ similarity matrix, i.e., $\{\tilde{a}_{uv}^{(l)}\}_{N \times N}$, required by Eqn. 2, thus also reducing the learning difficulties.

Nevertheless, Eqn. 5 still suffers what we mentioned the over-normalizing issue. The crux is that the message passing is operated on a weighted fully-connected graph where, in fact, only partial edges are important. Also, such a deterministic way of feature aggregation over all the instances may increase the risk for over-fitting, especially when $N$ is large. We next resolve the issues by distilling a sparse structure from the fully-connected graph.

**Differentiable Stochastic Structure Learning.** The difficultly lies in how to enable differentiable optimization for discrete graph structures. The weight $\tilde{a}_{uv}^{(l)}$ given by Eqn. 2 could be used to define a categorical distribution for generating latent edges from distribution $\text{Cat}(\boldsymbol{\pi}_u^{(l)})$ where $\boldsymbol{\pi}_u^{(l)} = \{\pi_{uv}^{(l)}\}_{v=1}^N$ and $\pi_{uv}^{(l)} = p(v|u) = \tilde{a}_{uv}^{(l)}$. Then in principle, we can sample over the categorical distribution multiple times for each node to obtain its neighbors. However, the sampling process would introduce discontinuity and hinders back-propagation. Fortunately, we notice that the Eqn. 3 can be modified to incorporate the reparametrization trick [16] to allow differentiable learning:

$$\mathbf{z}_u^{(l+1)} = \sum_{v=1}^N \frac{\exp((\mathbf{q}_u^\top\mathbf{k}_v + g_v)/\tau)}{\sum_{w=1}^N \exp((\mathbf{q}_u^\top\mathbf{k}_w + g_w)/\tau)} \cdot \mathbf{v}_v = \sum_{v=1}^N \frac{\kappa(\mathbf{q}_u/\sqrt{\tau}, \mathbf{k}_v/\sqrt{\tau})e^{g_v/\tau}}{\sum_{w=1}^N \kappa(\mathbf{q}_u/\sqrt{\tau}, \mathbf{k}_w/\sqrt{\tau})e^{g_w/\tau}} \cdot \mathbf{v}_v, \quad (6)$$

where $g_u$ is i.i.d. sampled from Gumbel distribution and $\tau$ is a temperature coefficient. Eqn. 6 is a continuous relaxation of sampling one neighbored node for $u$ over $\text{Cat}(\boldsymbol{\pi}_u^{(l)})$ and $\tau$ controls the closeness to hard discrete samples [23]. Following similar reasoning as Eqn. 3 and 5, we can yield

$$
\mathbf{z}_u^{(l+1)} \approx \sum_{v=1}^{N} \frac{\phi(\mathbf{q}_u/\sqrt{\tau})^\top \phi(\mathbf{k}_v/\sqrt{\tau}) e^{g_v/\tau}}{\sum_{w=1}^{N} \phi(\mathbf{q}_u/\sqrt{\tau})^\top \phi(\mathbf{k}_w/\sqrt{\tau}) e^{g_w/\tau}} \cdot \mathbf{v}_v = \frac{\phi(\mathbf{q}_u/\sqrt{\tau})^\top \sum_{v=1}^{N} e^{g_v/\tau} \phi(\mathbf{k}_v/\sqrt{\tau}) \cdot \mathbf{v}_v^\top}{\phi(\mathbf{q}_u/\sqrt{\tau})^\top \sum_{w=1}^{N} e^{g_w/\tau} \phi(\mathbf{k}_w/\sqrt{\tau})}.
\tag{7}
$$

Eqn. 7 achieves message passing over a sampled latent graph (where we only sample once for each node) and still guarantees linear complexity as Eqn. 5. In practice, we can sample $K$ times (e.g., $K = 5$) for each node and take an average of the aggregated results. Due to space limit, we defer more details concerning the differentiable sampling-based message passing to Appendix A. Besides, in Fig. 5 and Alg. 1 of Appendix A, we present an illustration for node embedding updating in each layer, from a matrix view that is practically used for implementation.

### 3.2 Well-posedness of the Kernelized Gumbel-Softmax Operator

One reasonable concern for Eqn. 7 is whether the RF approximation for kernel functions maintains the well-posedness of Gumbel approximation for the target discrete variables. As a justification for the new message-passing function, we next answer two theoretical questions: 1) How is the approximation capability of RF for the original dot-then-exponentiate operation with Gumbel variables in Eqn. 6? 2) Does Eqn. 7 still guarantee a continuous relaxation of the categorical distributions? We formulate the results as follows and defer proofs to Appendix B.

**Theorem 1** (Approximation Error for Softmax-Kernel). *Assume $\|\mathbf{q}_u\|_2$ and $\|\mathbf{k}_v\|_2$ are bounded by $r$, then with probability at least $1 - \epsilon$, the gap $\Delta = \left| \phi(\mathbf{q}_u/\sqrt{\tau})^\top \phi(\mathbf{k}_v/\sqrt{\tau}) - \kappa(\mathbf{q}_u/\sqrt{\tau}, \mathbf{k}_v/\sqrt{\tau}) \right|$, where $\phi$ is defined by Eqn. 4, will be bounded by $\mathcal{O}\left( \sqrt{\frac{\exp(6r/\tau)}{m\epsilon}} \right)$.*

We can see that the error bound of RF for approximating original softmax-kernel function depends on both the dimension of feature map $\phi$ and temperature $\tau$. Notably, the error bound is independent of node number $N$, which implies that the approximation ability is insensitive to dataset sizes.

The second question is non-trivial since Eqn. 7 involves randomness of Gumbel variables and random transformation in $\phi$, which *cannot* be decoupled apart. We define $c_{uv} = \frac{\phi(\mathbf{q}_u/\sqrt{\tau})^\top \phi(\mathbf{k}_v/\sqrt{\tau}) e^{g_v/\tau}}{\sum_{w=1}^{N} \phi(\mathbf{q}_u/\sqrt{\tau})^\top \phi(\mathbf{k}_w/\sqrt{\tau}) e^{g_w/\tau}}$ as the result from the kernelized Gumbel-Softmax and $\mathbf{c}_u = \{c_{uv}\}_{v=1}^{N}$ denotes the sampled edge vector for node $u$. We can arrive at the result as follows.

**Theorem 2** (Property of Kernelized Gumbel-Softmax Random Variables). *Suppose $m$ is sufficiently large, we have the convergence property for the kernelized Gumbel-Softmax operator*

$$
\lim_{\tau \to 0} \mathbb{P}(c_{uv} > c_{uv'}, \forall v' \neq v) = \frac{\exp(\mathbf{q}_u^\top \mathbf{k}_v)}{\sum_{w=1}^{N} \exp(\mathbf{q}_u^\top \mathbf{k}_w)}, \quad \lim_{\tau \to 0} \mathbb{P}(c_{uv} = 1) = \frac{\exp(\mathbf{q}_u^\top \mathbf{k}_v)}{\sum_{w=1}^{N} \exp(\mathbf{q}_u^\top \mathbf{k}_w)}.
$$

It shows that when i) the dimension of feature map is large enough and ii) the temperature goes to zero, the distribution from which latent structures are sampled would converge to the original categorical distribution.

*Remark.* The two theorems imply a trade-off between RF approximation and Gumbel-Softmax approximation w.r.t. the choice of $\tau$. A large $\tau$ would help to reduce the burden on kernel dimension $m$, and namely, small $\tau$ would require a very large $m$ to guarantee enough RF approximation precision. On the other hand, if $\tau$ is too large, the weight on each edge will converge to $\frac{1}{N}$, i.e., the model nearly degrades to mean pooling, while a small $\tau$ would endow the kernelized Gumbel-Softmax with better approximation to the categorical distribution. Empirical studies on this are presented in Appendix E.

### 3.3 Input Structures as Relational Bias

Eqn. 7 does not leverage any information from observed geometry which, however, is often recognized important for modeling physically-structured data [3]. We therefore accommodate input topology (if any) as relational bias via modifying the attention weight as $\tilde{a}_{uv}^{(l)} \leftarrow \tilde{a}_{uv}^{(l)} + \mathbb{I}[a_{uv} = 1]\sigma(b^{(l)})$,

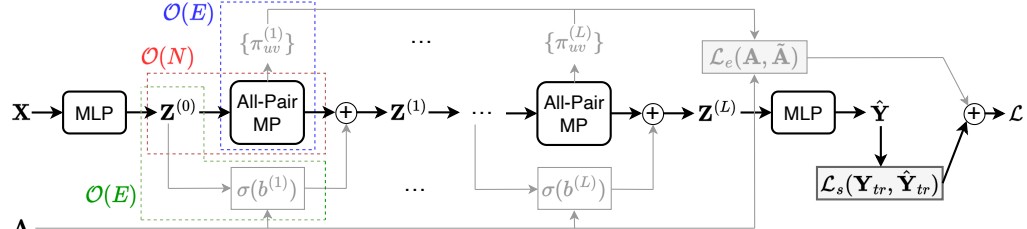

Figure 1: Illustration for the data flow of NODEFORMER which takes node embedding matrix $\mathbf{X}$ and (optional) graph adjacency matrix $\mathbf{A}$ as input. There are three components in NODEFORMER. The first one is the all-pair message passing (MP) module (colored red) which adopts our proposed kernelized Gumbel-Softmax operator to update node embeddings in each layer with $\mathcal{O}(N)$ complexity. The other two components are optional based on the availability of input graphs: 1) relational bias (colored green) that reinforces the propagation weight on observed edges; 2) edge regularization loss (colored blue) that aims to maximize the probability for observed edges. These two components require $\mathcal{O}(E)$ complexity. The final training loss $\mathcal{L}$ is the weighted sum of the standard supervised classification loss and the edge regularization loss.

where $b^{(l)}$ is a learnable scalar as relational bias for any adjacent node pairs $(u, v)$ and $\sigma$ is a certain (bounded) activation function like sigmoid. The relational bias aims at assigning adjacent nodes in $\mathcal{G}$ with proper weights, and the node representations could be accordingly updated by

$$\mathbf{z}_u^{(l+1)} \leftarrow \mathbf{z}_u^{(l+1)} + \sum_{v, a_{uv}=1} \sigma(b^{(l)}) \cdot \mathbf{v}_v. \tag{8}$$

Eqn. 8 increases the algorithmic complexity for message passing to $\mathcal{O}(N + E)$, albeit within the same order-of-magnitude as common GNNs operating on input graphs. Also, one can consider higher-order adjacency as relational bias for better expressiveness at some expense of efficiency, as similarly done by [1]. We summarize the feed-forward computation of NODEFORMER in Alg. 1.

### 3.4 Learning Objective

Given training labels $\mathbf{Y}_{tr} = \{y_u\}_{u \in \mathcal{N}_{tr}}$, where $\mathcal{N}_{tr}$ denotes the set of labeled nodes, the common practice is to maximize the observed data log-likelihood which yields a supervised loss (with $C$ classes)

$$\mathcal{L}_s(\mathbf{Y}_{tr}, \hat{\mathbf{Y}}_{tr}) = -\frac{1}{N_{tr}} \sum_{v \in \mathcal{N}_{tr}} \sum_{c=1}^{C} \mathbb{I}[y_u = c] \log \hat{y}_{u,c}, \tag{9}$$

where $\mathbb{I}[\cdot]$ is an indicator function. However, it may not suffice to generalize well due to that the graph topology learning increases the degrees of freedom and the number of training labels is not comparable to that. Therefore, we additionally introduce an edge-level regularization:

$$\mathcal{L}_e(\mathbf{A}, \tilde{\mathbf{A}}) = -\frac{1}{NL} \sum_{l=1}^{L} \sum_{(u,v) \in \mathcal{E}} \frac{1}{d_u} \log \pi_{uv}^{(l)}, \tag{10}$$

where $d_u$ denotes the in-degree of node $u$ and $\pi_{uv}^{(l)}$ is the predicted probability for edge $(u, v)$ at the $l$-th layer. Eqn. 10 is a maximum likelihood estimation for edges in $\mathcal{E}$, with data distribution defined

$$p_0(v|u) = \begin{cases} \frac{1}{d_u}, & a_{uv} = 1 \\ 0, & otherwise. \end{cases} \tag{11}$$

We next show how to efficiently obtain $\pi_{uv}^{(l)}$. Although the feed-forward NODEFORMER computation defined by Eqn. 7 does not explicitly produce the value for each $\pi_{uv}^{(l)}$, we can query their values by

$$\pi_{uv}^{(l)} = \frac{\phi(W_Q^{(l)} \mathbf{z}_u^{(l)})^\top \phi(W_K^{(l)} \mathbf{z}_v^{(l)})}{\phi(W_Q^{(l)} \mathbf{z}_u^{(l)})^\top \sum_{w=1}^{N} \phi(W_K^{(l)} \mathbf{z}_w^{(l)})}, \tag{12}$$

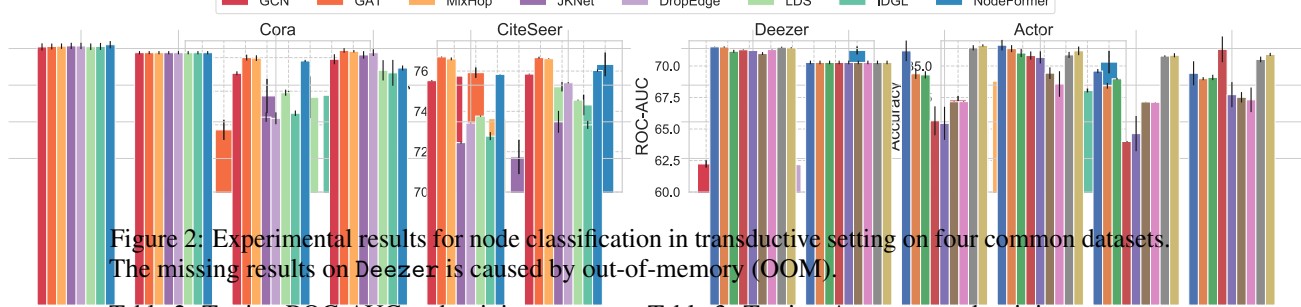

Figure 2: Experimental results for node classification in transductive setting on four common datasets. The missing results on `Deezer` is caused by out-of-memory (OOM).

Table 2: Testing ROC-AUC and training memory cost on `OGB-Proteins` with batch size 10K.

| Method | ROC-AUC (%) | Train Mem |
|---|---|---|
| MLP | $72.04 \pm 0.48$ | 2.0 GB |
| GCN | $72.51 \pm 0.35$ | 2.5 GB |
| SGC | $70.31 \pm 0.23$ | 1.2 GB |
| GraphSAINT-GCN | $73.51 \pm 1.31$ | 2.3 GB |
| GraphSAINT-GAT | $74.63 \pm 1.24$ | 5.2 GB |
| NODEFORMER | $\mathbf{77.45} \pm 1.15$ | 3.2 GB |
| NODEFORMER-dt | $75.50 \pm 0.64$ | 3.1 GB |
| NODEFORMER-tp | $76.18 \pm 0.09$ | 3.2 GB |

Table 3: Testing Accuracy and training memory cost on `Amazon2M` with batch size 100K.

| Method | Accuracy (%) | Train Mem |
|---|---|---|
| MLP | $63.46 \pm 0.10$ | 1.4 GB |
| GCN | $83.90 \pm 0.10$ | 5.7 GB |
| SGC | $81.21 \pm 0.12$ | 1.7 GB |
| GraphSAINT-GCN | $83.84 \pm 0.42$ | 2.1 GB |
| GraphSAINT-GAT | $85.17 \pm 0.32$ | 2.2 GB |
| NODEFORMER | $\mathbf{87.85} \pm 0.24$ | 4.0 GB |
| NODEFORMER-dt | $87.02 \pm 0.75$ | 2.9 GB |
| NODEFORMER-tp | $87.55 \pm 0.11$ | 4.0 GB |

where the summation term can be re-used from once computation, as is done by Eqn. 5 and Eqn. 7. Therefore, after once computation for the summation that requires $\mathcal{O}(N)$, the computation for each $\pi_{uv}^{(l)}$ requires $\mathcal{O}(1)$ complexity, yielding the total complexity controlled within $\mathcal{O}(E)$ (since we only need to query the observed edges). The final objective can be the combination of two: $\mathcal{L} = \mathcal{L}_s + \lambda \mathcal{L}_e$, where $\lambda$ controls how much emphasis is put on input topology. We depict the whole data flow of NODEFORMER's training in Fig. 1.

## 4 Evaluation

We consider a diverse set of datasets for experiments and present detailed dataset information in Appendix D. For implementation, we set $\sigma$ as sigmoid function and $\tau$ as 0.25 for all datasets. The output prediction layer is a one-layer MLP. More implementation details are presented in Appendix C. All experiments are conducted on a NVIDIA V100 with 16 GB memory.

As baseline models, we basically consider GCN [19] and GAT [36]. Besides, we compare with some advanced GNN models, including JKNet [44] and MixHop [1]. These GNN models all rely on input graphs. We further consider DropEdge [28] and two SOTA graph structure learning methods, LDS-GNN [11] and IDGL [4] for comparison. For large-scale datasets, we additionally compare with two scalable GNNs, a linear model SGC [39] and a graph-sampling model GraphSAINT [48]. More detailed information about these models are presented in Appendix C. All the experiments are repeated five times with different initializations.

### 4.1 Experiments on Transductive Node Classification

We study supervised node classification in transductive setting on common graph datasets: `Cora`, `Citeseer`, `Deezer` and `Actor`. The first two have high homophily ratios and the last two are identified as heterophilic graphs [53, 21]. These datasets are of small or medium sizes (with 2K~20K nodes). We use random splits with train/valid/test ratios as 50%/25%/25%. For evaluation metrics, we use ROC-AUC for binary classification on `Deezer` and Accuracy for other datasets with more than 2 classes. Results are plotted in Fig. 2 and NODEFORMER achieves the best mean Accuracy/ROC-AUC across four datasets and in particular, outperforms other models by a large margin on two heterophilic graphs. The results indicate that NODEFORMER can handle both homophilious and non-homophilious graphs. Compared with two structure learning models LDS and IDGL, NODEFORMER yields significantly better performance, which shows its superiority. Also, for `Deezer`, LDS and IDGL suffers from out-of-memory (OOM). In fact, the major difficulty for `Deezer` is the large

Table 4: Experimental results on semi-supervised classficiation on `Mini-ImageNet` and `20News-Groups` where we use $k$-NN (with different $k$'s) for artificially constructing an input graph.

| Method | Mini-ImageNet | | | | 20News-Group | | | |
|---|---|---|---|---|---|---|---|---|
| | $k=5$ | $k=10$ | $k=15$ | $k=20$ | $k=5$ | $k=10$ | $k=15$ | $k=20$ |
| GCN | $84.86 \pm 0.42$ | $85.61 \pm 0.40$ | $85.93 \pm 0.59$ | $85.96 \pm 0.66$ | $65.98 \pm 0.68$ | $64.13 \pm 0.88$ | $62.95 \pm 0.70$ | $62.59 \pm 0.62$ |
| GAT | $84.70 \pm 0.48$ | $85.24 \pm 0.42$ | $85.41 \pm 0.43$ | $85.37 \pm 0.51$ | $64.06 \pm 0.44$ | $62.51 \pm 0.71$ | $61.38 \pm 0.88$ | $60.80 \pm 0.59$ |
| DropEdge | $83.91 \pm 0.24$ | $85.35 \pm 0.44$ | $85.25 \pm 0.63$ | $85.81 \pm 0.65$ | $64.46 \pm 0.43$ | $64.01 \pm 0.42$ | $62.46 \pm 0.51$ | $62.68 \pm 0.71$ |
| IDGL | $83.63 \pm 0.32$ | $84.41 \pm 0.35$ | $85.50 \pm 0.24$ | $85.66 \pm 0.42$ | $65.09 \pm 1.23$ | $63.41 \pm 1.26$ | $61.57 \pm 0.52$ | $62.21 \pm 0.79$ |
| LDS | OOM | OOM | OOM | OOM | $66.15 \pm 0.36$ | $64.70 \pm 1.07$ | $63.51 \pm 0.64$ | $63.51 \pm 1.75$ |
| NODEFORMER | $86.77 \pm 0.45$ | $86.74 \pm 0.23$ | $86.87 \pm 0.41$ | $86.64 \pm 0.42$ | $66.01 \pm 1.18$ | $65.21 \pm 1.14$ | $64.69 \pm 1.31$ | $64.55 \pm 0.97$ |
| NODEFORMER w/o graph | $87.46 \pm 0.36$ | | | | $64.71 \pm 1.33$ | | | |

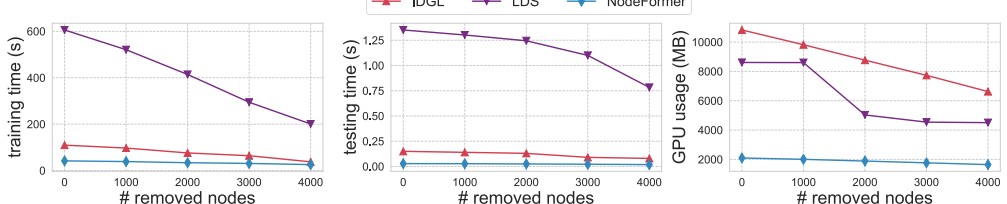

Figure 3: Comparison of training/inference time and GPU memory cost w.r.t. different instance numbers (by removing a certain portion of nodes) on `20News-Groups`.

dimensions of input node features (nearly 30K), which causes OOM for IDGL even with the anchor approximation. In contrast, NODEFORMER manages to scale and produce desirable accuracy.

## 4.2 Experiments on Larger Graph Datasets

To further test the scalability, we consider two large-sized networks, `OGB-Proteins` and `Amazon2M`, with over 0.1 million and 2 million of nodes, respectively. `OGB-Proteins` is a multi-task dataset with 112 output dimensions, while `Amazon2M` is extracted from the Amazon Co-Purchasing network that entails long-range dependence [13]. For `OGB-Proteins`, we use the protocol of [15] and ROC-AUC for evaluation. For `Amazon2M`, we adopt random splitting with 50%/25%/25% nodes for training, validation and testing, respectively. Due to the large dataset size, we adopt mini-batch partition for training, in which case, for NODEFORMER we only consider structure learning among nodes in a random mini-batch. We use batch size 10000 and 100000 for `Proteins` and `Amazon2M`, respectively. While the mini-batch partition may sacrifice the exposure to all instances, we found using large batch size can yield decent performance, which is also allowable thanks to the $\mathcal{O}(N)$ complexity of our model. For example, even setting the batch size as 100000, we found NODEFORMER costs only 4GB GPU memory for training on `Amazon2M`. Table 2 presents the results on `OGB-Proteins` where for fair comparison mini-batch training is also used for other models except GraphSAINT. We found that NODEFORMER yields much better ROC-AUC and only requires comparable memory as simple GNN models. Table 3 reports the results on `Amazon2M` which shows that NODEFORMER outperforms baselines by a large margin and the memory cost is even fewer than GCN. This shows its practical efficacy and scalability on large-scale datasets and also the capability for addressing long-range dependence with shallow layers (we use $L = 3$).

## 4.3 Experiments on Graph-Enhanced Applications

We apply our model to semi-supervised image and text classification on `Mini-ImageNet` and `20News-Groups` datasets, without input graphs. The instances of `Mini-ImageNet` [37] are 84×84 RGB images and we randomly choose 30 classes each of which contains 600 samples for experiments. `20News-Groups` [25] consists of nearly 10K texts whose features are extracted by TF-IDF. More details for preprocessing are presented in Appendix D. Also, for each dataset, we randomly split instances into 50%/25%/25% for train/valid/test. Since there is no input graph, we use $k$-NN (over input node features) for artificially constructing a graph for enabling GNN's message passing and the graph-based components (edge regularization and relational bias) of NODEFORMER. Table 4 presents the comparison results under different $k$'s. We can see that NODEFORMER achieves the best performance in seven cases out of eight. The performance of GNN competitors varies significantly with different $k$ values, and NODEFORMER is much less sensitive. Intriguingly, when we do not use

the input graph, i.e., removing both the edge regularization and relational bias, NODEFORMER can still yield competitive even superior results on `Mini-ImageNet`. This suggests that the $k$-NN graphs are not necessarily informative and besides, our model learns useful latent graph structures from data.

### 4.4 Further Discussions

**Comparison of Time/Space Consumption.** Fig. 3 plots training/inference time and GPU memory costs of NODEFORMER and two SOTA structure learning models. Compared with LDS, NODE-FORMER reduces the training time, inference time, memory cost by up to 93.1%, 97.9%, 75.6%, respectively; compared with IDGL (using anchor-based approximation for speedup), NODEFORMER reduces the training time, inference time, memory cost by up to 61.8%, 80.8%, 80.6%, respectively.

**Ablation on Stochastic Components.** Table 2 and 3 also include two variants of NODEFORMER for ablation study. 1) NODEFORMER-dt: replace Gumbel-Softmax by original Softmax (with temperature 1.0) for deterministic propagation; 2) NODEFORMER-tp: use original Softmax with temperature set as 0.25 (the same as NODEFORMER). There is performance drop when removing the Gumbel components, which may be due to over-normalizing or over-fitting that are amplified in large datasets, as we discussed in Section 3.1 and the kernelized Gumbel-Softmax operator shows its effectiveness.

**Ablation on Edge Loss and Relational Bias.** We study the effects of edge-level regularization and relation bias as ablation study shown in Table 6 located in Appendix E, where the results consistently show that both components contribute to some positive effects and suggest that our edge-level loss and relation bias can both help to leverage useful information from input graphs.

**Impact of Temperature and Feature Map Dimension.** We study the effects of $\tau$ and $m$ in Fig. 6 located in Appendix E and the variation trend accords with our theoretical analysis in Section 3.2. Specifically, the result shows that the test accuracy increases and then falls with the temperature changing from low to high values (usually achieves the peak accuracy with a temperature of 0.4). Besides, we can see that when the temperature is relatively small, the test accuracy goes high with the dimension of random features increasing. However, when the temperature is large, the accuracy would drop even with large feature dimension $m$. Such a phenomenon accords with the theoretical result presented in Section 3.2. For low temperature which enables desirable approximation performance for Gumbel-Softmax, then larger random feature dimension would help to produce better approximation to the original exponentiate-then-dot operator. In contrast, high temperature could not guarantee precise approximation for the original categorical distribution, which deteriorates the performance.

**Visualization and Implications.** Fig. 4 visualizes node embeddings and edge connections (filter out the edges with weights larger than a threshold) on `20News-Groups` and `Mini-Imagenet`, which show that NODEFORMER tends to assign more weights for nodes with the same class and sparse edges for nodes with different classes. This helps to interpret why NODEFORMER improves the performance on downstream node-level prediction: the latent structures can propagate useful information to help the model learn better node representations that can be easily distinguished by the classifier. We also compare the learned structures with original graphs in Fig. 7 located in Appendix E. We can see that the latent structures learned by NODEFORMER show different patterns from the observed ones, especially for heterophilic graphs. Another interesting phenomenon is that there exist some dominant nodes which are assigned large weights by other nodes, forming some vertical 'lines' in the heatmap. This suggests that these nodes could contain critical information for the learning tasks and play as pivots that could improve the connectivity of the whole system.

## 5 Why NODEFORMER Improves Downstream Prediction?

There remains a natural question concerning our learning process: how effective can the learned latent topology be for downstream tasks? We next dissect the rationale from a Bayesian perspective. In fact, our model induces a predictive distribution $p(\mathbf{Y}, \tilde{\mathbf{A}}|\mathbf{X}, \mathbf{A}) = p(\tilde{\mathbf{A}}|\mathbf{X}, \mathbf{A})p(\mathbf{Y}|\tilde{\mathbf{A}}, \mathbf{X}, \mathbf{A})$ where we can treat the estimated graph $\tilde{\mathbf{A}}$ as a latent variable.[2] Specifically, $p(\tilde{\mathbf{A}}|\mathbf{X}, \mathbf{A})$ is instantiated with the structure estimation module and $p(\mathbf{Y}|\tilde{\mathbf{A}}, \mathbf{X}, \mathbf{A})$ is instantiated with the feature propagation module. In principle, ideal latent graphs should account for downstream tasks and maximize the potentials

---

[2]We assume one latent graph to simplify the illustration though we practically learn layer-specific graphs for each layer of NODEFORMER. The analysis can be trivially extended to such a case.

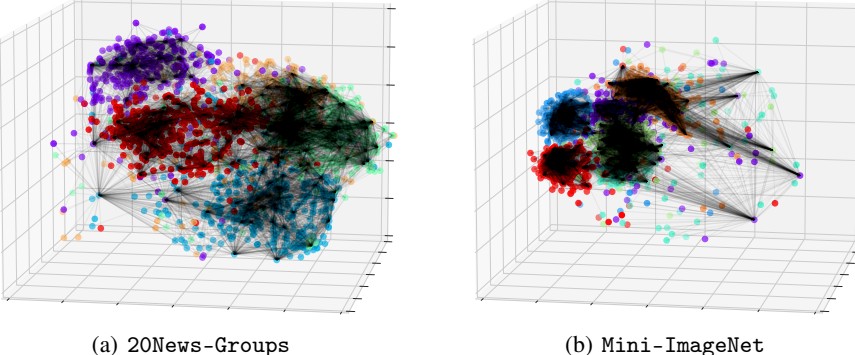

(a) `20News-Groups`           (b) `Mini-ImageNet`

Figure 4: Visualization of node embeddings and edge connections produced by NODEFORMER on graph-enhanced application datasets. We mark the nodes with a particular class with one color. More comparison between the learned structures and original input graphs is presented in Appendix E.

of message passing for producing informative node representations. Thus, optimal latent graphs presumably come from the posterior $p(\tilde{\mathbf{A}}|\mathbf{Y}, \mathbf{X}, \mathbf{A}) = \frac{p(\mathbf{Y}|\mathbf{X}, \mathbf{A}, \tilde{\mathbf{A}})p(\tilde{\mathbf{A}}|\mathbf{X}, \mathbf{A})}{\int_{\mathbf{Y}} p(\mathbf{Y}|\mathbf{X}, \mathbf{A}, \tilde{\mathbf{A}})p(\tilde{\mathbf{A}}|\mathbf{X}, \mathbf{A})d\mathbf{Y}}$ which is given by Bayes theorem. Unfortunately, such a posterior is unknown and intractable for the integration.

**A Variational Perspective.** An intriguing conclusion stems from another view into the learning process: we can treat the structure estimation as a variational distribution $q(\tilde{\mathbf{A}}|\mathbf{X}, \mathbf{A})$ and our learning objective in Section 3.4 can be viewed as the embodiment of a minimization problem over the predictive and variational distributions via

$$p^*, q^* = \arg\min_{p,q} \underbrace{-\mathbb{E}_q[\log p(\mathbf{Y}|\tilde{\mathbf{A}}, \mathbf{X}, \mathbf{A})]}_{\mathcal{L}_s} + \underbrace{\mathcal{D}(q(\tilde{\mathbf{A}}|\mathbf{X}, \mathbf{A})\|p_0(\tilde{\mathbf{A}}|\mathbf{X}, \mathbf{A}))}_{\mathcal{L}_e}, \qquad (13)$$

where $\mathcal{D}$ denotes the Kullback-Leibler divergence. Specifically, the *predictive* term is equivalent to minimizing the supervised loss (with Gumbel-Softmax as a surrogate for sampling-based estimates over $q(\tilde{\mathbf{A}}|\mathbf{X}, \mathbf{A})$), and the KL *regularization* term is embodied with the edge-level MLE loss (Eqn. 10) (if we define the prior distribution $p_0(\tilde{\mathbf{A}}|\mathbf{X}, \mathbf{A})$ following Eqn. 11). One may notice that Eqn. 13 is essentially the Evidence Lower Bound (ELBO) for the log-likelihood $\log p(\mathbf{Y}|\mathbf{X}, \mathbf{A})$.

**Proposition 1.** *Assume $q$ can exploit arbitrary distributions over $\tilde{\mathbf{A}}$. When Eqn. 13 achieves the optimum, we have 1) $\mathcal{D}(q(\tilde{\mathbf{A}}|\mathbf{X}, \mathbf{A})\|p(\tilde{\mathbf{A}}|\mathbf{Y}, \mathbf{X}, \mathbf{A})) = 0$ and 2) $\log p(\mathbf{Y}|\mathbf{X}, \mathbf{A})$ is maximized.*

The proposition indicates that our adopted learning objective intrinsically minimizes the divergence between latent graphs generated by the model and the samples from the posterior $p(\tilde{\mathbf{A}}|\mathbf{Y}, \mathbf{X}, \mathbf{A})$ that ideally helps to propagate useful adjacent information w.r.t. downstream tasks. Therefore, a well-trained network of NODEFORMER on labeled data could produce effective latent topology that contributes to boosting the downstream performance.

## 6 Conclusion

This paper proposes a scalable and efficient graph Transformer (especially for node level) that can propagate layer-wise node signals between arbitrary pairs beyond input topology. The key module, a kernelized Gumbel-Softmax operator, enables us to learn layer-specific latent graphs with linear algorithmic complexity without compromising the precision. The results on diverse graph datasets and situations verify the effectiveness, scalability, and stability. We provide more discussions on the limitations and potential impacts in Appendix F.

## Acknowledgement

This work was partly supported by National Key Research and Development Program of China (2020AAA0107600), National Natural Science Foundation of China (61972250, 72061127003), and Shanghai Municipal Science and Technology (Major) Project (22511105100, 2021SHZDZX0102).

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
