# Appendix

## A More Details for NODEFORMER

### A.1 Differentiable Sampling-based Message Passing on Latent Structures

We provide more details concerning the differentiable sampling-based message passing through our kernelized Gumbel-Softmax operator, as complementary to the content of Sec. 3.1. As illustrated in Sec. 3.1, the $l$-th layer's feature propagation is defined over the $l$-th layer's latent graph composed of the sampled edges $e_{uv}^{(l)} \sim \text{Cat}(\boldsymbol{\pi}_u)^{(l)}$. For each layer, we sample $K$ times for each node, i.e., there will be $K$ sampled neighbored nodes for each node $u$. We assume $\tilde{\mathcal{E}}^{(l)} = \{e_{uv}^{(l)}\}$ as the set of sampled edges in the latent graph of the $l$-th layer. Then the updating rule for node embeddings at the $l$-th layer based on the latent graph can be written as

$$\mathbf{z}_u^{(l+1)} = \frac{1}{K} \sum_{v, e_{uv}^{(l)} \in \tilde{\mathcal{E}}^{(l)}} \mathbf{v}_u = \frac{1}{K} \sum_v \mathbb{I}[e_{uv}^{(l)} \in \tilde{\mathcal{E}}^{(l)}] \mathbf{v}_u. \tag{14}$$

The above equation introduces dis-continuity due to the sampling process that disables the end-to-end differentiable training. We thus adopt Gumbel-Softmax as a reparameterization trick to approximate the discrete sampled results via continuous relaxation:

$$\mathbf{z}_u^{(l+1)} \approx \frac{1}{K} \sum_{k=1}^{K} \sum_{v=1}^{N} \frac{\exp((\mathbf{q}_u^\top \mathbf{k}_u + g_{kv})/\tau)}{\sum_{w=1}^{N} \exp((\mathbf{q}_u^\top \mathbf{k}_w + g_{kw})/\tau)} \cdot \mathbf{v}_u, \ g_{kw} \sim \text{Gumbel}(0,1). \tag{15}$$

The temperature $\tau$ controls the closeness to hard discrete samples [23]. If $\tau$ is close to zero, then the Gumbel-Softmax term $\frac{\exp((\mathbf{q}_u^\top \mathbf{k}_u + g_{kv})/\tau)}{\sum_{w=1}^{N} \exp((\mathbf{q}_u^\top \mathbf{k}_w + g_{kw})/\tau)}$ for any $v$ converges to a one-hot vector:

$$\frac{\exp((\mathbf{q}_u^\top \mathbf{k}_v + g_{kv})/\tau)}{\sum_{w=1}^{N} \exp((\mathbf{q}_u^\top \mathbf{k}_w + g_{kw})/\tau)} = \begin{cases} 1, & \text{if } v \text{ satisfies } \mathbf{q}_u^\top \mathbf{k}_v + g_{kv} > \mathbf{q}_u^\top \mathbf{k}_{v'} + g_{kv'} \forall v' \neq v, \\ 0, & otherwise. \end{cases} \tag{16}$$

The Eqn. 15 requires $\mathcal{O}(N^2)$ for computing the embeddings for $N$ nodes in one layer. To reduce the complexity to $\mathcal{O}(N)$, we resort to the kernel approximation idea, following similar reasoning as Eqn. 3 and 5:

$$\begin{aligned}
\mathbf{z}_u^{(l+1)} &\approx \frac{1}{K} \sum_{k=1}^{K} \sum_{v=1}^{N} \frac{\exp((\mathbf{q}_u^\top \mathbf{k}_u + g_{kv})/\tau)}{\sum_{w=1}^{N} \exp((\mathbf{q}_u^\top \mathbf{k}_w + g_{kw})/\tau)} \cdot \mathbf{v}_u \\
&= \frac{1}{K} \sum_{k=1}^{K} \sum_{v=1}^{N} \frac{\exp((\mathbf{q}_u^\top \mathbf{k}_u + g_{kv})/\tau)}{\sum_{w=1}^{N} \exp((\mathbf{q}_u^\top \mathbf{k}_w + g_{kw})/\tau)} \cdot \mathbf{v}_u \\
&= \frac{1}{K} \sum_{k=1}^{K} \sum_{v=1}^{N} \frac{\kappa(\mathbf{q}_u/\sqrt{\tau}, \mathbf{k}_v/\sqrt{\tau}) e^{g_{kv}/\tau}}{\sum_{w=1}^{N} \kappa(\mathbf{q}_u/\sqrt{\tau}, \mathbf{k}_w/\sqrt{\tau}) e^{g_{kw}/\tau}} \cdot \mathbf{v}_v \\
&\approx \frac{1}{K} \sum_{k=1}^{K} \sum_{v=1}^{N} \frac{\phi(\mathbf{q}_u/\sqrt{\tau})^\top \phi(\mathbf{k}_v/\sqrt{\tau}) e^{g_{kv}/\tau}}{\sum_{w=1}^{N} \phi(\mathbf{q}_u/\sqrt{\tau})^\top \phi(\mathbf{k}_w/\sqrt{\tau}) e^{g_{kw}/\tau}} \cdot \mathbf{v}_v \\
&= \frac{1}{K} \sum_{k=1}^{K} \frac{\phi(\mathbf{q}_u/\sqrt{\tau})^\top \sum_{v=1}^{N} e^{g_{kv}/\tau} \phi(\mathbf{k}_v/\sqrt{\tau}) \cdot \mathbf{v}_v^\top}{\phi(\mathbf{q}_u/\sqrt{\tau})^\top \sum_{w=1}^{N} e^{g_{kw}/\tau} \phi(\mathbf{k}_w/\sqrt{\tau})}.
\end{aligned} \tag{17}$$

The above result yields the one-layer updating rule for NODEFORMER's feed-forwarding w.r.t. each node $u$. In terms of practical implementation, we adopt matrix multiplications for computing the node embeddings for all the nodes in the next layer, for which we present the details in the next subsection.

### A.2 Model Implementation from the Matrix View

In practice, the implementation of NODEFORMER is based on matrix operations that simultanenously update all the nodes in one layer. We present the feed-forward process of NODEFORMER from a

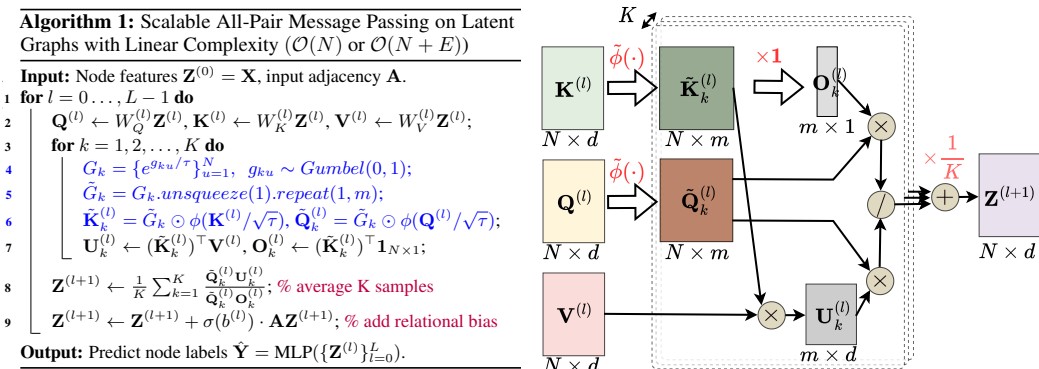

**Algorithm 1:** Scalable All-Pair Message Passing on Latent Graphs with Linear Complexity ($\mathcal{O}(N)$ or $\mathcal{O}(N+E)$)

**Input:** Node features $\mathbf{Z}^{(0)} = \mathbf{X}$, input adjacency $\mathbf{A}$.

1 **for** $l = 0 \ldots, L-1$ **do**
2      $\mathbf{Q}^{(l)} \leftarrow W_Q^{(l)}\mathbf{Z}^{(l)}$, $\mathbf{K}^{(l)} \leftarrow W_K^{(l)}\mathbf{Z}^{(l)}$, $\mathbf{V}^{(l)} \leftarrow W_V^{(l)}\mathbf{Z}^{(l)}$;
3      **for** $k = 1, 2, \ldots, K$ **do**
4          $G_k = \{e^{g_{ku}/\tau}\}_{u=1}^N$,   $g_{ku} \sim Gumbel(0,1)$;
5          $\tilde{G}_k = G_k.unsqueeze(1).repeat(1,m)$;
6          $\tilde{\mathbf{K}}_k^{(l)} = \tilde{G}_k \odot \phi(\mathbf{K}^{(l)}/\sqrt{\tau})$, $\tilde{\mathbf{Q}}_k^{(l)} = \tilde{G}_k \odot \phi(\mathbf{Q}^{(l)}/\sqrt{\tau})$;
7          $\mathbf{U}_k^{(l)} \leftarrow (\tilde{\mathbf{K}}_k^{(l)})^\top \mathbf{V}^{(l)}$, $\mathbf{O}_k^{(l)} \leftarrow (\tilde{\mathbf{K}}_k^{(l)})^\top \mathbf{1}_{N \times 1}$;
8      $\mathbf{Z}^{(l+1)} \leftarrow \frac{1}{K}\sum_{k=1}^K \frac{\tilde{\mathbf{Q}}_k^{(l)}\mathbf{U}_k^{(l)}}{\tilde{\mathbf{Q}}_k^{(l)}\mathbf{O}_k^{(l)}}$; % average K samples
9      $\mathbf{Z}^{(l+1)} \leftarrow \mathbf{Z}^{(l+1)} + \sigma(b^{(l)}) \cdot \mathbf{A}\mathbf{Z}^{(l+1)}$; % add relational bias

**Output:** Predict node labels $\hat{\mathbf{Y}} = \text{MLP}(\{\mathbf{Z}^{(l)}\}_{l=0}^L)$.

Figure 5: Alg. 1 presents the details for NODEFORMER's feed-forward process from a matrix view that is practically used in our implementation. The figure illustrates the layer-wise node representation updating based on the kernelized Gumbel-Softmax operator, which reduces the algorithmic complexity from quadratic to $\mathcal{O}(N)$ via avoiding explicit computation of the all-pair similarities. $\odot$ in Alg. 1 denotes element-wise product. $\tilde{\phi}(\cdot)$ in the figure represents the random feature map with Gumbel noise whose details are shown by the blue part of Alg. 1.

matrix view in Fig. 5 where Alg. 1 depicts how node embeddings are updated in each layer through our introduced kernelized Gumbel-Softmax message passing in Sec. 3.1. The right sub-figure illustrates the one layer's updating which only requires $\mathcal{O}(N)$ complexity by avoiding the cumbersome all-pair similarity matrix.

# B   Proof for Technical Results

## B.1   Proof for Theorem 1

To prove our theorem, we first introduce the following lemma given by the Lemma 2 in [6].

**Proposition 1.** *Denote a softmax kernel as $SM(\mathbf{x},\mathbf{y}) = \exp(\mathbf{x}^\top\mathbf{y})$. The Positive Random Features defined by Eqn. 4 for softmax-kernel estimation, i.e., $\widehat{SM}_m(\mathbf{x},\mathbf{y}) = \frac{1}{m}\sum_{i=1}^m[\exp(\mathbf{w}_i^\top\mathbf{x} - \frac{\|\mathbf{x}\|^2}{2})\exp(\mathbf{w}_i^\top\mathbf{y} - \frac{\|\mathbf{y}\|^2}{2})]$, has the mean and variance over $\mathbf{w} \sim \mathcal{N}(0, I_d)$ as*

$$\mathbb{E}_{\mathbf{w}}(\widehat{SM}_m(\mathbf{x},\mathbf{y})) = SM(\mathbf{x},\mathbf{y}) = \exp(\mathbf{x}^\top\mathbf{y}),$$
$$\mathbb{V}_{\mathbf{w}}(\widehat{SM}_m(\mathbf{x},\mathbf{y})) = \frac{1}{m}\exp(\|\mathbf{x}+\mathbf{y}\|^2)SM^2(\mathbf{x},\mathbf{y}) \tag{18}$$
$$(1 - \exp(-\|\mathbf{x}+\mathbf{y}\|^2)).$$

The lemma shows that the Positive Random Features can achieve unbiased approximation for the softmax kernel with a quantified variance.

Back to our main theorem, suppose the L2-norms of $\mathbf{q}_u$ and $\mathbf{k}_v$ are bounded by $r$, we can derive the probability using the Chebyshev's inequality:

$$\mathbb{P}\left(\Delta \leq \sqrt{\frac{\exp(6r/\tau)}{m\epsilon}}\right) \geq 1 - \frac{\mathbb{V}_{\mathbf{w}}(\widehat{\text{SM}}_m(\mathbf{q}_u/\sqrt{\tau}, \mathbf{k}_v/\sqrt{\tau}))}{\exp(6r/\tau)/m\epsilon} \tag{19}$$

where $\Delta = \left|\widehat{\text{SM}}_m(\mathbf{q}_u/\sqrt{\tau}, \mathbf{k}_v/\sqrt{\tau}) - \text{SM}(\mathbf{q}_u/\sqrt{\tau}, \mathbf{k}_v/\sqrt{\tau})\right|$ denotes the deviation of the kernel approximation. Using the result in Lemma 1, we can further obtain that the RHS of Eqn. 19 is no greater than

$$1 - \epsilon\exp\left(\|\frac{\mathbf{q}_u + \mathbf{k}_v}{\sqrt{\tau}}\|^2 + 2\frac{\mathbf{q}_u^\top\mathbf{k}_v}{\tau} - 6\frac{r}{\tau}\right). \tag{20}$$

Since $\|\frac{\mathbf{q}_u + \mathbf{k}_v}{\sqrt{\tau}}\|^2 \leq \frac{4r}{\tau}$ and $2\frac{\mathbf{q}_u^\top\mathbf{k}_v}{\tau} \leq \frac{2r}{\tau}$, we can achieve the stated result:

$$\mathbb{P}\left(\Delta \leq \sqrt{\frac{\exp(6r/\tau)}{m\epsilon}}\right) \geq 1 - \epsilon. \tag{21}$$

## B.2 Proof for Theorem 2

Before entering the proof for the theorem, we first introduce two basic technical lemmas. While such results are already mentioned in previous studies [16, 23], their proofs will be useful for the subsequent reasoning. Therefore, we restate the proofs as building blocks for the following presentation.

**Proposition 2.** *Given real numbers $x_i, x_j \in \mathbb{R}$ and $u_i, u_j$ i.i.d. sampled from uniform distribution within $(0,1)$. With Gumbel perturbation defined as $g(u) = -\log(-\log(u))$, we have the probability*

$$P(x_i + g(u_i) > x_j + g(u_j)) = \frac{1}{1 + \exp(-(x_i - x_j))}.$$

*Proof.* Due to $g(u) = -\log(-\log(u))$, the inequality of interests $x_i + g(u_i) > x_j + g(u_j)$ can be rearranged as

$$e^{x_i - x_j} > \frac{\log(u_i)}{\log(u_j)}. \tag{22}$$

Since $\log(u_j) < 0$, Eqn. 22 can be written as

$$u_j < u_i^{e^{x_j - x_i}}. \tag{23}$$

As $u_i, u_j$ are i.i.d. sampled from a uniform distribution, the probability when the above formula can be calculated via:

$$\int_0^1 \int_0^{u_i^{e^{x_j - x_i}}} du_j du_i = \int_0^1 u_i^{e^{x_j - x_i}} du_i \tag{24}$$

$$= \frac{1}{1 + \exp(-(x_i - x_j))}.$$

Thus, we conclude the proof with

$$P(x_i + g(u_i) > x_j + g(u_j)) = \frac{1}{1 + \exp(-(x_i - x_j))}. \tag{25}$$

$\square$

**Proposition 3.** *Let $X \sim Gumbel(\alpha, \tau)$ (i.e. $X_k = \frac{\exp((\log \alpha_k + g_k)/\tau)}{\sum_{i=1}^n \exp((\log \alpha_i + g_i)/\tau)}$) with location parameters $\alpha \in (0, \infty)^n$ and temperature $\tau \in (0, \infty)$, then:*

- $P(X_k > X_i, \forall i \neq k) = \frac{\alpha_k}{\sum_{i=1}^n \alpha_i}$,

- $P(\lim_{\tau \to 0} X_k = 1) = \frac{\alpha_k}{\sum_{i=1}^n \alpha_i}$.

*Proof.* This result can be similarly proved as Lemma 2. The event of interests $X_k > X_i, \forall i \neq k$ is equivalent to

$$\log \alpha_k - \log(-\log u_k) > \log \alpha_1 - \log(-\log u_1),$$
$$\log \alpha_k - \log(-\log u_k) > \log \alpha_2 - \log(-\log u_2),$$
$$... \tag{26}$$
$$\log \alpha_k - \log(-\log u_k) > \log \alpha_n - \log(-\log u_n).$$

Since all the above inequalities are independent given $u_k$, we can rearrange the first inequality as

$$u_1 < u_k^{\alpha_1/\alpha_k} \leq 1. \tag{27}$$

Since $u_1 \sim U[0,1]$, the probability for the first inequality in Eqn. 26 being true would be $u_k^{\alpha_1/\alpha_k}$. Thus, the probability for Eqn. 26 being true can be calculated via

$$u_k^{\alpha_1/\alpha_k} u_k^{\alpha_2/\alpha_k} ... g_k^{\alpha_n/\alpha_k} = g_k^{(\alpha_1 + \alpha_2 + ... + \alpha_n)/\alpha_k} = g_k^{(1/\alpha_k) - 1}. \tag{28}$$

For simplicity, we assume $\sum_{i=1}^n \alpha_i = 1$. Then for any $g_k \in [0,1]$, we obtain

$$P(X_k > X_i, \forall i \neq k) = \int_0^1 g_k^{(1/\alpha_k) - 1} dg_k \tag{29}$$

$$= \frac{\alpha_k}{\sum_{i=1}^n \alpha_i},$$

and arrive at the result for the first bullet point. For the second bullet point, when $\tau \to 0$, we have

$$\lim_{\tau \to 0} \frac{\exp((\log \alpha_i + g_i)/\tau)}{\exp((\log \alpha_j + g_j)/\tau)}$$
$$= \lim_{\tau \to 0} \exp((\log \alpha_i + g_i - \log \alpha_j - g_j)/\tau) \tag{30}$$
$$= \begin{cases} \infty, & \text{if } \alpha_i > \alpha_j \\ 0, & otherwise. \end{cases}$$

Such a fact indicates that the output of a Concrete distribution with $\tau \to 0$ will be a one-hot vector ($X_{\arg\max_i \alpha_i} = 1$). This yields the conclusion that

$$P(\lim_{\tau \to 0} X_k = 1) = P(X_k > X_i, \forall i \neq k) = \frac{\alpha_k}{\sum_{i=1}^{n} \alpha_i}. \tag{31}$$

$\square$

Now we turn to the proof of our theorem. We are to prove that the kernelized form in Eqn. 7 has the same property as the original Gumbel-Softmax in the limit sense (when $\tau$ goes to zero). We recall that we have defined $\mathbf{q}_u = W_Q^{(l)} \mathbf{z}_u^{(l)}$, $\mathbf{k}_u = W_K^{(l)} \mathbf{z}_u^{(l)}$ and $\mathbf{v}_u = W_V^{(l)} \mathbf{z}_u^{(l)}$ for simplicity.

First, by definition we have

$$\phi(\frac{\mathbf{q}_u}{\sqrt{\tau}})^{\top} \phi(\frac{\mathbf{k}_v}{\sqrt{\tau}}) e^{\frac{g_v}{\tau}}$$
$$= \frac{1}{m} \exp(-\frac{||\frac{\mathbf{q}_u}{\sqrt{\tau}}||^2 + ||\frac{\mathbf{k}_v}{\sqrt{\tau}}||^2}{2}) \sum_{i=1}^{m} \exp(\omega_i^{\top}(\frac{\mathbf{q}_u}{\sqrt{\tau}} + \frac{\mathbf{k}_v}{\sqrt{\tau}}) + \frac{g_v}{\tau}). \tag{32}$$

The property holds that for $\forall w \neq v$, we have $\lim_{\tau \to 0} \frac{\phi(\frac{\mathbf{q}_u}{\sqrt{\tau}})^{\top} \phi(\frac{\mathbf{k}_v}{\sqrt{\tau}}) e^{\frac{g_v}{\tau}}}{\phi(\frac{\mathbf{q}_u}{\sqrt{\tau}})^{\top} \phi(\frac{\mathbf{k}_w}{\sqrt{\tau}}) e^{\frac{g_w}{\tau}}}$ equals to $\infty$ or 0, i.e. the output of the kernelized Gumbel-Softmax is still a one-hot vector when $\tau \to 0$. Let

$$Y_v = \frac{\phi(\frac{\mathbf{q}_u}{\sqrt{\tau}})^{\top} \phi(\frac{\mathbf{k}_v}{\sqrt{\tau}}) e^{\frac{g_v}{\tau}}}{\sum_{w=1}^{N} \phi(\frac{\mathbf{q}_u}{\sqrt{\tau}})^{\top} \phi(\frac{\mathbf{k}_w}{\sqrt{\tau}}) e^{\frac{g_w}{\tau}}}. \tag{33}$$

Here $Y_v$ is defined in the same way as $c_{uv}$ in Section 3.2. We thus have $P(\lim_{\tau \to 0} Y_v = 1) = P(Y_v > Y_{v'}, \forall v' \neq v)$.

To compute $P(Y_v > Y_{v'}, \forall v' \neq v)$, for simplicity, let us consider the probability $P(Y_v > Y_{v'}) = P(\phi(\frac{\mathbf{q}_u}{\sqrt{\tau}})^{\top} \phi(\frac{\mathbf{k}_v}{\sqrt{\tau}}) e^{\frac{g_v}{\tau}} > \phi(\frac{\mathbf{q}_u}{\sqrt{\tau}})^{\top} \phi(\frac{\mathbf{k}_{v'}}{\sqrt{\tau}}) e^{\frac{g_{v'}}{\tau}})$. To keep notation clean, we define

$$\beta_v = \phi(\frac{\mathbf{q}_u}{\sqrt{\tau}})^{\top} \phi(\frac{\mathbf{k}_v}{\sqrt{\tau}}), \ \beta_{v'} = \phi(\frac{\mathbf{q}_u}{\sqrt{\tau}})^{\top} \phi(\frac{\mathbf{k}_{v'}}{\sqrt{\tau}}). \tag{34}$$

Then the above-mentioned probability can be rewritten as $P(\log \beta_v + \frac{g_v}{\tau} > \log \beta_{v'} + \frac{g_{v'}}{\tau})$, where $\beta_v$ and $\beta_{v'}$ are two i.i.d. random variables.

From Lemma 1, we have $\mathrm{E}(\beta_v) = \exp(\mathbf{q}_u^{\top} \mathbf{k}_v / \tau) = \alpha_v^{\frac{1}{\tau}}$, $\mathrm{E}(\beta_{v'}) = \exp(\mathbf{q}_u^{\top} \mathbf{k}_{v'} / \tau) = \alpha_{v'}^{\frac{1}{\tau}}$, where $\alpha_v$ and $\alpha_{v'}$ are two constant values. Then using Lemma 2, we have

$$P(\log \alpha_v^{1/\tau} + \frac{g_v}{\tau} > \log \alpha_{v'}^{1/\tau} + \frac{g_{v'}}{\tau})$$
$$= P(\log \alpha_v + g_v > \log \alpha_{v'} + g_{v'})$$
$$= \frac{1}{1 + \exp(\log \alpha_{v'} - \log \alpha_v)} \tag{35}$$
$$= \frac{\alpha_{v'}}{\alpha_v + \alpha_{v'}}.$$

According to the Chebyshev's inequality, we have $P(|\beta_v - \alpha_v^{\frac{1}{\tau}}| \leq \epsilon_v) \geq 1 - \frac{\sigma_v^2}{\epsilon_v^2}$. Here $\sigma_v^2 = \mathbb{V}_{\mathbf{w}}(\widehat{\mathrm{SM}}_m(\frac{\mathbf{q}_u}{\sqrt{\tau}}, \frac{\mathbf{k}_j}{\sqrt{\tau}}))$, which can given by Lemma 1.

Due to the convexity of logarithmic function, we have

$$\frac{|\log \beta_v - \frac{1}{\tau} \log \alpha_v|}{|\beta_v - \alpha_v^{\frac{1}{\tau}}|} \leq \frac{1}{\alpha_v^{\frac{1}{\tau}} - \epsilon_v}, \tag{36}$$

and subsequently,

$$|\log \beta_v - \frac{1}{\tau} \log \alpha_v| \leq \frac{|\beta_v - \alpha_v^{\frac{1}{\tau}}|}{\alpha_v^{\frac{1}{\tau}} - \epsilon_v} \tag{37}$$

$$\leq \frac{\epsilon_v}{\alpha_v^{\frac{1}{\tau}} - \epsilon_v}.$$

Therefore we have $P(|\log \beta_v - \frac{1}{\tau} \log \alpha_v| \leq \frac{\epsilon_v}{\alpha_v^{\frac{1}{\tau}} - \epsilon_v}) \geq P(|\beta_v - \alpha_v^{\frac{1}{\tau}}| \leq \epsilon_v)$. Based on this, we can derive the result:

$$P(|\log \beta_v - \frac{1}{\tau} \log \alpha_v| \leq \frac{\epsilon_v}{\alpha_v^{\frac{1}{\tau}} - \epsilon_v}) \geq 1 - \frac{\sigma_v^2}{\epsilon_v^2}. \tag{38}$$

Since $\beta_v$ and $\beta_{v'}$ are two i.i.d. random variables, we have

$$P(|\log \beta_v - \frac{1}{\tau} \log \alpha_v| \leq \frac{\epsilon_v}{\alpha_v^{\frac{1}{\tau}} - \epsilon_v},$$

$$|\log \beta_{v'} - \frac{1}{\tau} \log \alpha_{v'}| \leq \frac{\epsilon_{v'}}{\alpha_{v'}^{\frac{1}{\tau}} - \epsilon_{v'}}) \geq (1 - \frac{\sigma_v^2}{\epsilon_v^2})(1 - \frac{\sigma_{v'}^2}{\epsilon_{v'}^2}). \tag{39}$$

For simplicity, we denote $\epsilon = \frac{\epsilon_v}{\alpha_v^{\frac{1}{\tau}} - \epsilon_v} + \frac{\epsilon_{v'}}{\alpha_{v'}^{\frac{1}{\tau}} - \epsilon_{v'}}$ and $P_\epsilon = (1 - \frac{\sigma_v^2}{\epsilon_v^2})(1 - \frac{\sigma_{v'}^2}{\epsilon_{v'}^2})$. We therefore have

$$P(|\log \beta_v - \frac{1}{\tau} \log \alpha_v| + |\log \beta_{v'} - \frac{1}{\tau} \log \alpha_{v'}| \leq \epsilon) \geq P_\epsilon. \tag{40}$$

Using the triangular inequality, we can yield

$$|(\log \beta_v - \frac{1}{\tau} \log \alpha_v) - (\log \beta_{v'} - \frac{1}{\tau} \log \alpha_{v'})|$$

$$\leq |\log \beta_v - \frac{1}{\tau} \log \alpha_v| + |\log \beta_{v'} - \frac{1}{\tau} \log \alpha_{v'}|. \tag{41}$$

Combining Eqn. 40 and 41, we have

$$P(|(\log \beta_v - \frac{1}{\tau} \log \alpha_v) - (\log \beta_{v'} - \frac{1}{\tau} \log \alpha_{v'})| \leq \epsilon) \geq P_\epsilon. \tag{42}$$

Let $c = \log \beta_v - \log \beta_{v'}$, so that $\mathrm{E}(c) = \frac{1}{\tau}(\log \alpha_v - \log \alpha_{v'})$. From Eqn. 42, we can obtain

$$P(c \geq \mathrm{E}(c) - \epsilon) \geq P_\epsilon. \tag{43}$$

According to Lemma 2, the probability $P(\mathrm{E}(c) - \epsilon \geq \frac{g_{v'} - g_v}{\tau})$ can be written as

$$P(\log \alpha_v - \log \alpha_{v'} - \tau\epsilon \geq g_{v'} - g_v)$$

$$= \frac{1}{1 + \exp(\log \alpha_{v'} - \log \alpha_v + \tau\epsilon)} \tag{44}$$

$$= \frac{1}{1 + \frac{\alpha_{v'}}{\alpha_v} e^{\tau\epsilon}}.$$

Since $c, g_v, g_{v'}$ are generated independently, combining Eqn. 43 and 44, we can yield

$$P(c \geq \frac{g_{v'} - g_v}{\tau}) \geq \frac{P_\epsilon}{1 + \frac{\alpha_{v'}}{\alpha_v} e^{\tau\epsilon}}. \tag{45}$$

Similarly, from Eq. 42 we have $P(c \leq \mathrm{E}(c) + \epsilon) \geq P_\epsilon$ and subsequently,

$$P(\frac{g_{v'} - g_v}{\tau} \geq \mathrm{E}(c) + \epsilon) = 1 - P(c + \epsilon \geq \frac{g_{v'} - g_v}{\tau})$$

$$= 1 - \frac{1}{1 + \frac{\alpha_{v'}}{\alpha_v} e^{-\tau\epsilon}}. \tag{46}$$

Thus we have $P(c \leq \frac{g_{v'}-g_v}{\tau}) \geq P_\epsilon(1 - \frac{1}{1+\frac{\alpha_{v'}}{\alpha_v}e^{-\tau\epsilon}})$ and also

$$P(c \geq \frac{g_{v'}-g_v}{\tau}) \leq 1 - P_\epsilon(1 - \frac{1}{1+\frac{\alpha_{v'}}{\alpha_v}e^{-\tau\epsilon}}). \tag{47}$$

Combining Eqn. 45 and 47, we conclude that

$$\frac{P_\epsilon}{1+\frac{\alpha_{v'}}{\alpha_v}e^{\tau\epsilon}} \leq P(c \geq \frac{g_{v'}-g_v}{\tau}) \leq 1 - P_\epsilon(1 - \frac{1}{1+\frac{\alpha_{v'}}{\alpha_v}e^{-\tau\epsilon}}). \tag{48}$$

Based on this we consider the limitation for two sides and thus obtain

$$\lim_{P_\epsilon \to 1}\lim_{\tau \to 0} P(c \geq \frac{g_{v'}-g_v}{\tau}) = \frac{1}{1+\frac{\alpha_{v'}}{\alpha_v}} = \frac{\alpha_v}{\alpha_v + \alpha_{v'}}. \tag{49}$$

Then with similar reasoning as Lemma 3, we have

$$\lim_{P_\epsilon \to 1}\lim_{\tau \to 0} P(Y_v = 1)$$
$$= \lim_{P_\epsilon \to 1}\lim_{\tau \to 0} P(Y_v > Y_{v'}, \forall v' \neq v) = \alpha_v/(\sum_{w=1}^{N} \alpha_w). \tag{50}$$

Recall that

$$P_\epsilon = (1 - \frac{\sigma_v^2}{\epsilon_v^2})(1 - \frac{\sigma_{v'}^2}{\epsilon_{v'}^2})$$
$$\sigma^2 = \mathbb{V}_{\mathbf{w}}(\widehat{SM}_m^+(\mathbf{x}, \mathbf{y})) \tag{51}$$
$$= \frac{1}{m}\exp(-\frac{\|\mathbf{x}\|^2 + \|\mathbf{y}\|^2}{2})\sum_{i=1}^{m}\exp(\mathbf{w}_i^\top(\mathbf{x}+\mathbf{y})),$$

where $\mathbf{x} = \frac{\mathbf{q}_u}{\sqrt{\tau}}, \mathbf{y} = \frac{\mathbf{k}_{v,v'}}{\sqrt{\tau}}$. Therefore, $P_\epsilon$ is dependent of the precision $\epsilon_v, \epsilon_{v'}$, the random feature dimension $m$, and the temperature $\tau$. If $m$ is sufficiently large, $\sigma$ would converge to zero and $P_\epsilon$ goes to 1. In such a case, Eqn. 50 holds once $\tau$ tends to zero. We thus conclude the proof.

## B.3 Proof for Proposition 1

According to our definitions in Section 5, we have

$$\mathcal{D}_{KL}(q_\phi(\tilde{\mathbf{A}}|\mathbf{X}, \mathbf{A})\|p(\tilde{\mathbf{A}}|\mathbf{Y}, \mathbf{X}, \mathbf{A}))$$
$$= \int_{A^*} q_\phi(\tilde{\mathbf{A}}|\mathbf{X}, \mathbf{A})\log\frac{q_\phi(\tilde{\mathbf{A}}|\mathbf{X}, \mathbf{A})}{p(\tilde{\mathbf{A}}|\mathbf{Y}, \mathbf{X}, \mathbf{A})}d\tilde{\mathbf{A}}$$
$$= \int_{A^*} q_\phi(\tilde{\mathbf{A}}|\mathbf{X}, \mathbf{A})\log\frac{q_\phi(\tilde{\mathbf{A}}|\mathbf{X}, \mathbf{A})p_\theta(\mathbf{Y}|\mathbf{X}, \mathbf{A})}{p(\tilde{\mathbf{A}}, \mathbf{Y}|\mathbf{X}, \mathbf{A})}d\tilde{\mathbf{A}}$$
$$= \int_{A^*} q_\phi(\tilde{\mathbf{A}}|\mathbf{X}, \mathbf{A})\log\frac{q_\phi(\tilde{\mathbf{A}}|\mathbf{X}, \mathbf{A})p_\theta(\mathbf{Y}|\mathbf{X}, \mathbf{A})}{p(\tilde{\mathbf{A}}, \mathbf{Y}|\mathbf{X}, \mathbf{A})}d\tilde{\mathbf{A}} \tag{52}$$
$$= \int_{A^*} q_\phi(\tilde{\mathbf{A}}|\mathbf{X}, \mathbf{A})\log\frac{q_\phi(\tilde{\mathbf{A}}|\mathbf{X}, \mathbf{A})p_\theta(\mathbf{Y}|\mathbf{X}, \mathbf{A})}{p(\mathbf{Y}|\tilde{\mathbf{A}}, \mathbf{X}, \mathbf{A})p(\tilde{\mathbf{A}}|\mathbf{X}, \mathbf{A})}d\tilde{\mathbf{A}}$$
$$= -\mathbb{E}_{q_\phi(\tilde{\mathbf{A}}|\mathbf{X}, \mathbf{A})}[\log p(\mathbf{Y}|\tilde{\mathbf{A}}, \mathbf{X}, \mathbf{A})] + \log p_\theta(\mathbf{Y}|\mathbf{X}, \mathbf{A}) + \mathcal{D}_{KL}(q_\phi(\tilde{\mathbf{A}}|\mathbf{X}, \mathbf{A})\|p(\tilde{\mathbf{A}}|\mathbf{X}, \mathbf{A}))$$
$$= -\text{ELBO}(\theta, \phi) + \log p_\theta(\mathbf{Y}|\mathbf{X}, \mathbf{A})$$

Since we assume $q_\phi$ can exploit arbitrary distributions over $\tilde{\mathbf{A}}$, we know that when the ELBO is optimized to the optimum, $\mathcal{D}_{KL}(q_\phi(\tilde{\mathbf{A}}|\mathbf{X}, \mathbf{A})\|p(\tilde{\mathbf{A}}|\mathbf{Y}, \mathbf{X}, \mathbf{A})) = 0$ holds. Otherwise, there exists $\phi^* \neq \phi$ such that $\text{ELBO}(\theta, \phi^*) > \text{ELBO}(\theta, \phi)$. Pushing further, when the optimum is achieved, $\log p_\theta(\mathbf{Y}|\mathbf{X}, \mathbf{A})$ would equal to ELBO and namely is maximized.

# C  Implementation Details

We present implementation details in our experiments for reproducibility. For more concrete details concerning architectures and hyper-parameter settings for NODEFORMER, one can directly refer to our public repository https://github.com/qitianwu/NodeFormer. We next present descriptions for baseline models' implementation. For baseline models MLP, GCN, GAT, MixHop and JKnet, we use the implementation provided by [21][3]. For DropEdge and two structure learning baseline models (LDS and IDGL), we also refer to their implementation provided by the original papers [28, 11, 4]. Concretely, we use GCN as the backbone for them.

## C.1  Details for Node Classification Experiments in Sec. 4.1

**Architectures.** For experiments on the datasets Cora, Citeseer, Deezer and Actor, the baseline models (GCN, GAT, MixHop, JKNet) are implemented with the following settings:

- Two GNN layers with hidden size 32 by default (unless otherwise mentioned). GAT uses 8 attention heads followed by its original setting.

- The activation function is ReLU (except GAT using ELU).

The architecture of our NODEFORMER is specified as follows:

- Two message-passing layers with hidden size 32. We also consider multi-head designs for our all-pair attentive message passing, and for each head we use independent parameterization. The results for different heads are combined in an average manner in each layer.

- The activation function is ELU that is only used for input MLP, and we do not use any activation for the feature propagation layers. In terms of relational bias, we specify $\sigma$ as sigmoid function and consider 2-order adjacency to strengthen the observed links of the input graph.

**Training Details.** In each epoch, we feed the whole data into the model, calculate the loss and conduct gradient descent accordingly. Concretely, we use BCE-Loss for two-class classification and NLL-Loss for multi-class classification, the Adam optimizer is used for gradient-based optimization. The training procedure will repeat the above process until a given budget of 1000 epochs. Finally, we report the test accuracy achieved by the epoch that gives the highest accuracy on validation dataset.

**Hyperparameters.** For each model, we use grid search on validation set for hyper-parameter setting. The learning rate is searched within $\{0.01, 0.001, 0.0001, 0.00001\}$, weight decay within $\{0.05, 0.005, 0.0005, 0.00005\}$, and dropout probability within $\{0.0, 0.5\}$. The hyper-parameters for NODEFORMER is provided in our public codes. The hyperparameters for baseline models are set as follows (we use the same notation as the original papers).

- For GCN and GAT, the learning rate is 0.01, and weight decay is set to 0.05. No dropout is used.

- For MixHop, the hyperparameters are the same as above, except that we further use grid search for hidden channels within {8, 16, 32, 64, 128}. We adopt 2 hops for all the four datasets.

- For JKNet, GCN is used as the backbone. Learning rate is set to 0.01 for Deezer and 0.001 for all the other three datasets. We concatenate the features in the final stage for Deezer, while we use max-pooling for the three other datasets. The hidden size is set as default, except for Deezer as 64.

- For DropEdge, the hidden size is chosen from {32, 64, 96, 128, 160}, the learning rate is within {0.01, 0.001, 0.0001}, and the dropedge rate is chosen from {0.3, 0.4, 0.5}.

- For LDS, the sampling time $S = 16$, the patience window size $\rho = 6$, the hidden size $\in \{8, 16, 32, 64\}$, the inner learning rate $\gamma \in \{1e\text{-}4, 1e\text{-}3, 1e\text{-}2, 1e\text{-}1\}$, and the number of updates used to compute the truncated hypergradient $\tau \in \{5, 10, 15\}$.

- For IDGL, we use its original version without anchor approximation on Cora, Citeseer and Actor. For Deezer, even using anchor approximation, it would also suffer from out-of-memory. Besides, we set: $\epsilon = 0.01$, hidden size $\in \{16, 64, 96, 128\}$, $\lambda \in \{0.5, 0.6, 0.7, 0.8\}$, $\eta \in \{0, 0.1, 0.2\}$, $\alpha \in \{0, 0.1, 0.2\}$, $\beta \in \{0, 0.1\}$, $\gamma \in \{0.1, 0.2\}$, $m \in \{6, 9, 12\}$.

---

[3]https://github.com/CUAI/Non-Homophily-Benchmarks.

## C.2 Details for Node Classification on Larger Graphs in Sec. 4.2

**Architectures.** For experiments on the two large datasets (OGB-Proteins and Amazon2M), the baseline models are implemented with the following settings:

- Three GNN layers with hidden size 64.
- The activation function is ReLU (except GAT using ELU).

The architecture of our NODEFORMER is specified as follows:

- Three message-passing layers with hidden size 64. The head number is set as 1.
- The activation function is ELU that is used for all the layers. In terms of relational bias, we specify $\sigma$ as identity function and consider 1-order adjacency to strengthen the observed links of the input graph.

**Training Details.** In each epoch, we use random mini-batch partition to split the whole set of nodes and feed each mini-batch of nodes into the model for all-pair propagation, as we mentioned in Section 4.1. Similarly, we use BCE-Loss for two-class classification and NLL-Loss for multi-class classification, the Adam optimizer is used for gradient-based optimization. The training procedure will repeat the above process until a given budget of 1000 epochs. The evaluation on testing data is conducted on CPU which enables full-batch feature propagation. Finally, we report the test accuracy achieved by the epoch that gives the highest accuracy on validation dataset.

## C.3 Details for Graph-Enhanced Experiments in Sec. 4.3

**Architectures.** The architectures of baselines (GCN, GAT, LDS and IDGL) and NODEFORMER model are the same as the transductive setting, except that we use grid search to adaptively tune the hidden size. Besides, we also adopt BatchNorm for baseline models.

**Training Details.** The input data have no graph structures in this setting. As mentioned in Section 4.3, we use $k$-NN for artificially constructing a graph to enable message passing. The training procedure is the same as the transductive setting.

**Hyperparameters.** The hyperparameters for baseline models are listed as follows.

- For GCN, the learning rate is 0.01 and the weight decay is 5e-4 on both datasets. The size of hidden channel is set to 64. Dropout is not used during training.
- For GAT, the learning rate is 0.001 and the weight decay is 5e-3 on both datasets. The size of hidden channel is 32 on `Mini-ImageNet` and 64 on `20News`. No dropout is used during training. The number of attention heads is 8.
- For LDS, its hyperparameters are determined by grid search in the same manner as in the transductive setting.
- For IDGL, we use its anchor-based version that can scale to these two datasets. Besides, on `20News`, we set: hidden size=64, learning rate=0.01 $\lambda = 0.7$, $\eta = 0.1$, $\alpha = 0.1$, $\beta = 0$, $\gamma = 0.1$, $\epsilon = 0.01$, $m = 12$. On `Mini-ImageNet`, we set: hidden size=96, learning rate=0.01 $\lambda = 0.8$, $\eta = 0.2$, $\alpha = 0$, $\beta = 0$, $\gamma = 0.1$, $\epsilon = 0.01$, $m = 12$.

# D Dataset Information

We present detailed information for our used datasets concerning the data collection, preprocessing and statistic information. Table 5 provides an overview of the datasets we used in the experiment.

## D.1 Dataset Information

**Node Classification Datasets.** For experiments on transductive node classification, we evaluate our model on two homophilous datasets `Cora` and `Citeseer` [33], and other two non-homophilous datasets `Actor` [26] and `Deezer` [29]. The first two are citation network datasets that contain sparse bag-of-words feature vectors for each document and a list of citation links between documents. The citation links are treated as (undirected) edges and each document has a class label. `Deezer` is a social network of users on Deezer from European countries, where edges represent mutual follower relationships. The node features are based on artists liked by each user and nodes are labeled

Table 5: Information for experiment datasets.

| Dataset | Context | Property | # Task | # Nodes | # Edges | # Node Feat | # Class |
|---|---|---|---|---|---|---|---|
| Cora | Citation network | homophilous | 1 | 2,708 | 5,429 | 1,433 | 7 |
| Citeseer | Citation network | homophilous | 1 | 3,327 | 4,732 | 3,703 | 6 |
| Deezer | Social network | non-homophilous | 1 | 28,281 | 92,752 | 31,241 | 2 |
| Actor | Actors in movies | non-homophilous | 1 | 7,600 | 29,926 | 931 | 5 |
| OGB-Proteins | Protein interaction | multi-task classification | 112 | 132,534 | 39,561,252 | 8 | 2 |
| Amazon2M | Product co-occurrence | long-range dependence | 1 | 2,449,029 | 61,859,140 | 100 | 47 |
| Mini-ImageNet | Image classification | no graph/$k$-NN graph | 1 | 18,000 | 0 | 128 | 30 |
| 20News-Groups | Text classification | no graph/$k$-NN graph | 1 | 9,607 | 0 | 236 | 10 |

with reported gender. `Actor` is a graph representing actor co-occurrence in Wikipedia pages. Each node corresponds to an actor, and the edge between two nodes denotes co-occurrence on the same Wikipedia page. Node features correspond to some keywords in the Wikipedia pages and the nodes are classified into five categories w.r.t. words of actor's Wikipedia. These four datasets are relatively small with thousands of instances and edges (`Deezer` is the largest one with nearly 20K nodes).

To evaluate NODEFORMER's scalability, we further consider two large datasets: `OGB-Proteins` [15] and `Amazon2M` [5]. These two datasets have million-level nodes and edges and require the model for scalable training. The `OGB-Proteins` dataset is an undirected, and typed (according to species) graph in which nodes represent proteins and edges indicate different types of biologically meaningful associations between proteins. All edges come with 8-dimensional features, where each dimension represents the approximate confidence of a single association type and takes on values between 0 and 1. The proteins come from 8 species and our task is to predict the presence of 112 protein functions in a multi-label binary classification setup respectively. `Amazon2M` is extracted from Amazon Co-Purchasing network [24], where each node represents a product, and the graph link represents whether two products are purchased together, the node features are generated by extracting bag-of-word features from the product descriptions. The top-level categories are used as labels for the products.

**Graph-enhanced Application Datasets.** We evaluate our model on two datasets without graph structure: `20News-Groups` [25] and `Mini-ImageNet` [37]. The `20News` dataset is a collection of approximately 20,000 newsgroup documents (nodes), partitioned (nearly) evenly across 20 different newsgroups. We take 10 classes from 20 newsgroups and use words (TFIDF) with a frequency of more than 5% as features. The `Mini-ImageNet` dataset consists of 84×84 RGB images from 100 different classes with 600 samples per class. For our experiment use, we choose 30 classes from the dataset, each with 600 images (nodes) that have 128 features extracted by CNN.

## D.2 Dataset Preprocessing

All the datasets we used in the experiment are directly collected from the source, except `Mini-ImageNet`, whose features are extracted by ourselves. Following the setting of [31], we compute node embeddings via a CNN model with 4 convolutional layers followed by a fully-connected layer resulting in a 128 dimensional embedding. Finally, the 128 dimensional outputs are treated as the features of the nodes (images) for subsequent GNN-based downstream task.

## E  More Experiment Results

We present extra ablation study results on the four transductive datasets for NODEFORMER w/ and w/o relational bias and edge-level regularization. Fig. 6 studies the impact of the temperature $\tau$ and the dimension of feature map $m$ on `Cora`. Furthermore, we visualize the attention maps of two model layers and compare with original input graphs of `Cora`, `Citeseer`, `Deezer` and `Actor` in Fig. 7.

Table 6: Ablation study results on transductive datasets, where "rb" denotes relational bias and "reg" represents edge-level regularization.

| Dataset | NODEFORMER | NODEFORMER w/o reg | NODEFORMER w/o rb |
|---|---|---|---|
| Cora | **88.69** ± 0.46 | 81.98 ± 0.46 | 88.06 ± 0.59 |
| Citeseer | **76.33** ± 0.59 | 70.60 ± 1.20 | 74.12 ± 0.64 |
| Deezer | **71.24** ± 0.32 | 71.22 ± 0.32 | 71.10 ± 0.36 |
| Actor | **35.31** ± 1.29 | 35.15 ± 1.32 | 34.60 ± 1.32 |

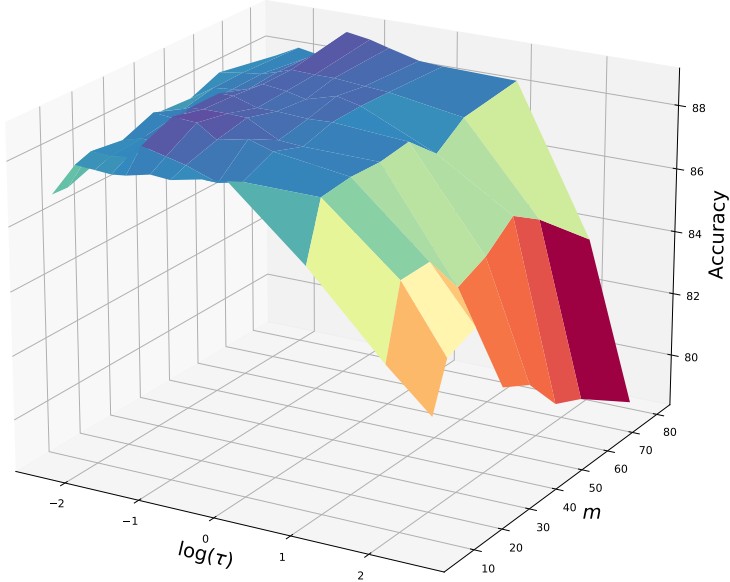

Figure 6: Impact of the temperature $\tau$ and the dimension of random feature map $m$ on Cora.

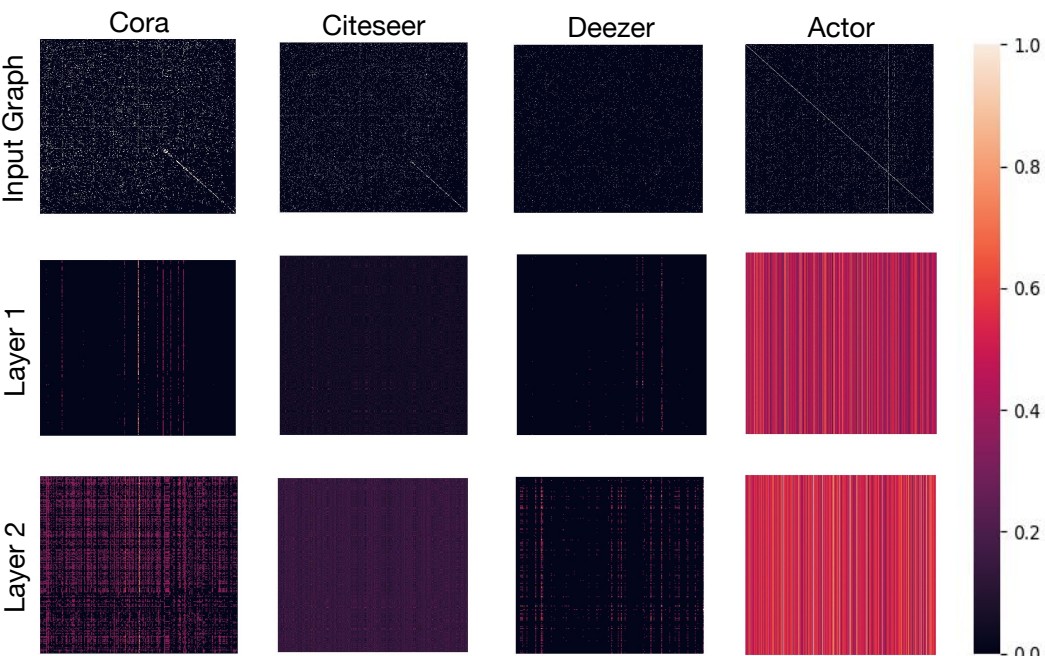

Figure 7: Visualization for input graph structures and latent graph structures (given by two layers of NODEFORMER) with colors reflecting the weights.

# F  Current Limitations, Outlooks and Potential Impacts

**Current Limitations.** In the present work, we focus on node classification for experiments, though NODEFORMER can be used as a flexible (graph) encoder for other graph-related problems such as graph classification, link prediction, etc. Beyond testing accuracy, social aspects like robustness and explainability can also be considered as the target for future works on top of NODEFORMER.

**Potential Impact.** Besides facilitating better node representations via message passing, graph structure learning also plays as key components in many other perpendicular problems in graph learning community, like explainability [47], knowledge transfer and distillation [45], adversarial robustness [50], training acceleration [34], handling feature extrapolation [40] and cold-start users in recommendation [41]. NODEFORMER can serve as a plug-in scalable structure learning encoder for uncovering underlying dependence, identifying novel structures and purifying noisy data in practical systems. Another promising direction is to leverage our kernelized Gumbel-Softmax operator as a plug-in module for designing efficient and expressive Transformers on graph data where the large graph size plays a critical performance bottleneck.