# OpenReview forum: "NodeFormer: A Scalable Graph Structure Learning Transformer for Node Classification"
_NeurIPS.cc/2022/Conference — NeurIPS 2022 Accept_

### Official Review · Reviewer_bJ9T · 2022-07-09

**Rating:** 5
**Confidence:** 4
**Soundness:** 3 good
**Presentation:** 3 good
**Contribution:** 2 fair

**Summary:**

This work performs message passing over any pairs of nodes in the graph, while avoiding the N^2 time complexity by using kernelized operator. The key technique is to use random feature to estimate the kernel function. The authors conduct experiments on various node-level tasks to evaluate the method.

**Questions:**

(1) For the edge loss (Eq. (10)), can we consider both observed edges and non-observed relationships. If we only consider existing edges, the loss could lead to all \pi_{uv} to be large.

**Limitations:**

Not applicable.

**Strengths And Weaknesses:**

#### Pros ####
(1) Using random feature technique to approximate the kernel function is impressive and reduces the complexity significantly.

(2) The presentation flow of this paper is clear and easy to follow.

#### Cons ####
(1) More convincing benchmark comparison should be considered. Actually, I was surprised by the effectiveness of the random feature approximation. I think more rigorous benchmark comparison should be used to evaluate the empirical performance. In the current experiments, except ogb-proteins, other experiments are mainly using the setting defined by this paper, which might be less convincing to me. For example, for the citation networks, we have well-established benchmark setup in the community. The setup used in this paper is different, and the results of the baselines are not consistent with prior works. For example, on Citeseer, the accuracy of JKNet (~72) is much lower than GCN (~76), which indicates that the reproduced results of baselines might not be convincing.




+++++++++ Post-rebuttal +++++++++

After reading the response from authors, my original concerns are addressed.

---

> ### Author Response · Authors · 2022-08-02
> **Response to Reviewer bJ9T**
>
> Thanks for your time and constructive feedback. The reviewer's main concerns primarily relate to aspects of our empirical evaluation, namely, consistency with the experimental design and baselines used in prior work.  To this end, we have provided new experimental results and attendant explanations that address these concerns.
>
> **Q1: More empirical results and comparison**
>
> We agree that it is generally better to use well-established settings when available or matched to the goal at hand. For ogb-proteins, as the reviewer noted, we already follow the benchmark setting for evaluation. For small networks, we originally used 50%/25%/25% random splitting for train/valid/test since we thought the main target scenario of our work is the situation where there is ample training data for learning effective latent structures. To make our evaluation more convincing, we add new experiments on the citation networks Cora and Citeseer following the splitting by [1], which is more widely used as standard benchmark by recent papers e.g., [2,3,4]. The results are shown below.
>
> |     | GCN | GAT | JKNet | MixHop | DropEdge | LDS | IDGL | ALIGNS |
> |------------|-------|-------|-------|-------|-------|-------|-------|-------|
> | Cora   | 85.89 ± 1.25 | 86.78 ± 1.67 | 86.21 ± 1.76  | 87.06 ± 0.97 | 85.12 ± 2.02  | 87.22 ± 2.01 | 87.53 ± 1.87 | 88.58 ± 1.21 |
> | Citeseer | 75.38 ± 1.61 | 75.87 ± 1.82 | 74.31 ± 1.89 | 75.12 ± 1.76 |75.12 ± 1.53 |  76.11 ± 1.91 |76.51 ± 1.87 | 77.21 ± 1.19
>
> We found that ALIGNS consistently outperforms the competitors, which further validates the superiority of the proposed approach.
>
>
> **Q2: Implementation of the baselines and the results of JKNet**
>
> As mentioned in our paper, we execute all competing models using public implementations, such as the ones provided by the Pytorch Geometric package (if available) or code provided with original papers. And for each model, we consider a hyper-parameter grid search using the search space reported in Appendix B. We believe such an evaluation is fair for comparison and comprehensive for reproducibility. Additionally, as for the JKNet result the reviewer mentioned, we noticed that the original paper [5] used different settings (Specifically, the splitting used by [5] is different from either the above common splitting or the splitting used in our paper). Therefore, with different data splitting, it is not surprising that the relative performances may be somewhat different. Indeed, one can see such differences in the results reproduced by other highly-cited papers with different splitting (we quote the scores from their papers):
> - Table 5 of [2]: GCN 76.68, JKNet 74.51
> - Table 2 of [3]: GCN 67.30, JKNet 60.85
> - Table 2 of [6]: GCN 75.40, JKNet 73.03
> - Table 2 of [4]): GCN 71.1, JKNet 69.8
>
> From the above, we observe that the performance of JKNet is inferior (about 1-6 points lower) than GCN on Citeseer. We thus believe that our reproduced results are convincing and consistent with existing works. Even so, thanks to the reviewer for properly considering the importance of consistency checks in evaluating empirical models; this is frequently overlooked.
>
> **Q3: If we only consider existing edges, the loss could lead to all $\pi_{uv}$ to be large.**
>
> This is not generally a concern for the following reason. In our model, the computation of $\pi_{uv}$ is originally defined (Eqn. 2) and further approximated (Eqn. 12) via
> $$\pi_{uv}^{(l)} = \frac{\exp((W_Q^{(l)} \mathbf z_u^{(l)})^\top (W_K^{(l)} \mathbf z_v^{(l)}))}{\sum_{w=1}^N \exp((W_Q^{(l)} \mathbf z_u^{(l)})^\top (W_K^{(l)} \mathbf z_w^{(l)}))} \approx \frac{\phi(W_Q^{(l)} \mathbf z_u^{(l)})^\top  \phi(W_K^{(l)} \mathbf z_v^{(l)}) }{ \phi(W_Q^{(l)} \mathbf z_u^{(l)})^\top \sum_{w=1}^N \phi(W_K^{(l)} \mathbf z_w^{(l)})}. $$
> One can see that the denominator involves the estimated similarities for all the other nodes and associates $\pi_{uv}^{(l)}$ with other $\pi_{uv'}^{(l)}$ ($v'\neq v$). Therefore, when we maximize the MLE loss (Eqn. 10) over observed edges, there will be two effects: 1) increasing the probabilities for observed edges (by updating gradient from the numerator of $\pi_{uv}^{(l)}$); 2) decreasing the probabilities for unobserved edges (by updating gradient from the denominator of $\pi_{uv}^{(l)}$). In short, the edge loss Eqn. 10 inherently involves both positive examples (observed edges) and negative examples (unobserved edges) through the Softmax functions.
>
>
> References:
>
> [1] GEOM-GCN - Geometric Graph Convolutional Networks, ICLR2020
>
> [2] Beyond Homophily in Graph Neural Networks- Current Limitations and Effective Designs, NeurIPS2020
>
> [3] ADAPTIVE UNIVERSAL GENERALIZED PAGERANK GRAPH NEURAL NETWORK, ICLR 2021
>
> [4] Simple and Deep Graph Convolutional Networks, ICML 2020
>
> [5] Representation Learning on Graphs with Jumping Knowledge Networks, ICML 2018
>
> [6] PREDICT THEN PROPAGATE: GRAPH NEURAL NETWORKS MEET PERSONALIZED PAGERANK, ICLR 2019

---

> > ### Comment · Reviewer_bJ9T · 2022-08-02
> > **Thanks for the response**
> >
> > Thank the authors for the response. My original concerns are addressed. I raised my score accordingly.

---

### Official Review · Reviewer_Xhvj · 2022-07-10

**Rating:** 7
**Confidence:** 3
**Soundness:** 4 excellent
**Presentation:** 4 excellent
**Contribution:** 4 excellent

**Summary:**

This paper considers to overcome the deficiencies of over-squashing, heterophily, long-range dependencies, edge incompleteness etc. by proposing a new structure to learn the topology of the input graphs. Specifically, it proposes to incorporate the learned topology into the attention matrix through the Softmax Gumbel operator. To reduce the runtime complexity, a kernelized version of attention computation is developed using random feature decomposition. Also, convergence properties of the approximation are derived. Extensive experiments are conducted on different types of node classification tasks.

**Questions:**

1. In Section 3.4, there is a comment stating that Eqn. (12) can be computed in $\mathcal{O}(1)$ because the summation term is computed in Eqn. (7). However, it seems that the summation terms are
\begin{equation}
\sum_{v=1}^{N} e^{g_v / \tau} \phi( \mathbf{k}_v / \sqrt{\tau})  \cdot  \mathbf{v}_v, \text{and}
\end{equation}

\begin{equation}
\sum_{w=1}^{N} e^{g_w / \tau} \phi( \mathbf{k}_w / \sqrt{\tau})
\end{equation}

instead of  $\sum_{w = 1}^{N} \phi(\mathbf{k}_w)$ . Could the authors elaborate more about this?


2. According to the error bound for Softmax Gumbel RF decomposition in theorem 1, the bound grows exponentially in terms of the length of embedded vector $q_u$ and $k_v$, but shrinks polynomially in terms of the number of random features. Have the authors measured the gap between the approximation and the truth value when working on the experiments? As the norm grows, is a large number of random features required?


**Limitations:**

The limitations and negative social impacts are not addressed in the paper.

**Strengths And Weaknesses:**

Strengths:

1. Theoretical analysis is comprehensive. Convergence results are developed to study the quality of the fast computation of the Softmax Gumbel operator.
2. The variational perspective of the proposed method is motivating.
3. Numerical experiments are comprehensive as well, which includes testing on small graphs, large graphs, transductive and inductive task, and graph-enhanced applications etc.

Weaknesses:

1. The paper mentions that the new method is proposed to tackle problems like over-squashing, heterophily, long-range dependencies etc. However, in the main body of the paper, it is hard to realize how the development of the new structure helps to resolve these issues accordingly. It would be great if the authors could point this out more clearly.
2. Since the method is proposed for scalable GNN, the authors are encouraged to review and discuss more about SOTA scalable GNNs and compare with the proposed model in the related works section, such as subgraph-wise sampling: GraphSAINT [1]; Kernelized masked attention: GKAT [2]; Linearized GNN: [3][4] etc. Also, those models could be reasonable baselines in the Section 4.2, Experiments on Larger Graph Datasets, where currently only MLP and GCN are compared.

[1] Zeng, Hanqing, et al. "Graphsaint: Graph sampling based inductive learning method." arXiv preprint arXiv:1907.04931 (2019).

[2] Choromanski, Krzysztof, et al. "Graph kernel attention transformers." arXiv preprint arXiv:2107.07999 (2021).

[3] Wu, Felix, et al. "Simplifying graph convolutional networks." International conference on machine learning. PMLR, 2019.

[4] Bojchevski, Aleksandar, et al. "Scaling graph neural networks with approximate pagerank." Proceedings of the 26th ACM SIGKDD International Conference on Knowledge Discovery & Data Mining. 2020.

---

> ### Author Response · Authors · 2022-08-02
> **Response to Reviewer Xhvj**
>
> Thank you for your time and thorough comments.
>
> **Q1: How does our approach handle long-range dependence, over-squashing, heterophily?**
>
> A common feature of GNN models is the recursive aggregation of node features/representations along existing graph structures, which leads to the potential for over-squashing and a deficiency for handling long-range dependencies and heterophily. We provide detailed elaboration of these issues as follows:
>
> ● Long-range dependence [1]: Since GNN models can only aggregate information from connected nodes in each layer, the effective receptive field is limited to local neighbors. For graphs with long-range dependencies where the information from remote nodes would be helpful for prediction on the centered nodes, GNNs need to stack many layers to bridge the information path between two nodes. By contrast, our model ALIGNS enables potential feature propagation between arbitrary pairs of nodes in each layer, which can easily capture long-range dependencies when necessary.
>
> ● Over-squashing: With GNNs the information shared across nodes can sometimes become overly squashed and exponentially dilated through multi-layer graph convolution [2]. Such a pheonomenon is related to the geometric property of input graph structure where the existing 'bottleneck' is the crux of the issue. Our model ALIGNS can potentially mitigate this phenomena by the all-pair attentions adaptively learning propagation strengthes between any node pairs, i.e., pursue optimal topology in each layer to improve the information utility.
>
> ● Heterophily: GNNs may yield undesirable results when the input graphs tend to connect nodes with distinct labels, i.e., heterophily graphs [3]. In these situations, message passing on the fixed input structures may propagate inconsistent information. However, in our model the latent structures are adaptively learned by node representations towards minimizing the prediction loss, which can guide the model to yield appropriate message-passing structures.
>
> **Q2: More comparison with scalable GNN models.**
>
> Thank you for sharing these related works; we will add a discussion of them to the related work section in final version where we have an extra page. In short, the technical aspect of our model is orthogonal to these scalable GNNs: we focus on developing scalable message passing on (all-pair) latent graphs, while they focus on either neighbor sampling or light-weighted feature propagation on (fixed) observed graphs.
>
> We also conducted new experiments using SGC and GraphSAINT as baselines for comparison on Proteins and Amazon2M. The results are shown below, where our ALIGNS outperforms them by significant margins:
>
> |     | SGC | GraphSAINT-GCN | GraphSAINT-GAT | ALIGNS |
> |------------|-------|-------|-------|-------|
> | Proteins   | 70.31 ± 0.23 | 73.51 ± 1.31 | 74.63 ± 1.24  | 77.45  ± 1.15  |
> | Amazon2M |  81.21 ± 0.12   |  83.84 ± 0.42 | 85.17 ± 0.32 | 87.85 ± 0.24 |
>
> We will add these comparison results to the final version; good suggestion, this definitely helps to strengthen our paper.
>
> **Q3: Should the summation term involve the Gumbel noise $e^{g_v/\tau}$ ?**
>
> The referenced summation term from Eqn 12 should not involve the Gumbel noise. Eqn. 12 aims to compute estimation for the probability of the edge between nodes v and u, i.e., $\pi_{uv}$, so it is based on the kernelized approximation for the original softmax attention score. In Eqns 6 and 7, the Gumbel noise is introduced as an approximation for sampling over the distribution defined by such edge probability $\pi_{uv}$ to obtain discrete graph structures. Differently, Eqn. 12 only needs to obtain the edge probability instead of the sampled result, and so does not need the Gumbel term.
>
> **Q4: What is the impact of input feature norm and random feature dimension on the approximation power?**
>
> Actually the norms of $q$ and $k$ are controlled by Layer Normalization used in each layer (the LN is used to reduce the vector norms before the all-pair propagation layer), so one can treat the norms as a constant when it comes to the kernelized approximation and the approximation error is mainly dependent on temperature $\tau$ and random feature dimension $m$. We did experiments on synthetic datasets where we fixed the attention scores as ground-truth and used the approximation results to compute the error gap; in doing so we found that the error is decreasing w.r.t. $\tau$ and $m$, which is consistent with our theoretical results. Also, we have provided a thorough study of the impact of $\tau$ and $m$ on downstream performance in Fig. 4 in the appendix originally submitted (see Appendix D.2 for discussions).
>
> **Q5: The limitations and negative social impacts are not addressed in the paper.**
>
> We have addressed the limitations and social impacts in Section E of Appendix in the originally uploaded version.

---

> > ### Author Response · Authors · 2022-08-02
> > **References for the previous response**
> >
> > References
> >
> > [1] Learning steady-states of iterative algorithms over graphs, ICML2018
> >
> > [2] On the bottleneck of graph neural networks and its practical implications, ICLR2021
> >
> > [3] Beyond Homophily in Graph Neural Networks- Current Limitations and Effective Designs, NeurIPS2020

---

> > > ### Comment · Reviewer_Xhvj · 2022-08-06
> > > **Regarding Q3 in the response**
> > >
> > > Thanks for the authors’ responses. Most of the responses address my concern. Look forward to seeing these adjustments in the future version.
> > >
> > > However, I still have a question to Q3. Sorry for not being clear enough. Actually what I originally meant is that as mentioned in the Sec. 3.4,  Eqn. 12 can be computed in O(1) because the summation term can be re-used from Eqn. 7. However, in Eqn. 7, all the summation terms involve the Gumbel noise $e^{(g_v/\tau)}$. My question is that how the summation term in Eqn. 12 can be re-used from computation of Eqn. 7?

---

> > > > ### Author Response · Authors · 2022-08-06
> > > > **Further clarification for Q3**
> > > >
> > > > Thank you for the valuable feedback and pointing out the part we can further improve.
> > > >
> > > > We have modified the description under Eqn. 12 in a more precise way in the newly uploaded paper (see the part colored blue). Actually, the summation term can be re-used from once (extra) computation, as is similarly done by Eqn. 5 and 7. This is exactly what we did in our code. After once computation for the summation $\sum_{w=1}^N \phi(W_K^{(l)} \mathbf z_w^{(l)})$ that requires $\mathcal O(N)$, then each $\pi_{uv}^{(l)}$ can be computed within $\mathcal O(1)$, yielding total complexity $\mathcal O(E)$.

---

> > > > > ### Comment · Reviewer_Xhvj · 2022-08-06
> > > > > **Thanks for clarification**
> > > > >
> > > > > Thanks for the authors' further clarification. This makes sense to me now. I will raise my score to 7.

---

### Official Review · Reviewer_N5pw · 2022-07-10

**Rating:** 7
**Confidence:** 3
**Soundness:** 4 excellent
**Presentation:** 3 good
**Contribution:** 3 good

**Summary:**

The authors proposed a GNN framework named ALIGNS that can simultaneously learn the latent topology and missing labels of nodes. They came up with a two steps approximation to first kernelize the softmax function used in latent node structures, then a continues relaxation of the categorical distribution over all pairs of edges to enable end-to-end back propagation. In the experimentation section they tested on various datasets with up to 2M nodes, in both transductive and inductive setting, given clean, noisy, and even no graph information. Results show not only ALIGNS has a performance gain against SOTA models, but also it scales with increasing number of nodes.

**Questions:**

- I feel like the same approximation can be applied to GAT or other attention based methods. What is the benefits on doing it for all pair-wise then using existing structures? Are we missing out on informations we already have in this case?
- Why is k-NN still needed in Graph-enhanced applications? Is observed adjacency strictly needed or we can use the current layer's latent structure as adjacency?
- Again in Eq(8) on the right side, is \z and \v of the same dimension? Is there no dimension changes during feature transformation?

**Limitations:**

I didn't see a negative societal impact of this work.

**Strengths And Weaknesses:**

Pros:
- The paper is well-written and easy to read.
- Method has potential for used in much wider range without constraining on graph-structured input.
- Scalability of model is very impressive. The kernelization and continues relaxation seems to hold up pretty well in real world task.
- Performance in transductive setting with standard benchmarks in homophily and heterophily graphs are on par or outperform SOTA.
- In inductive setting even after adding noise to edges, the model still performs very stable.

Cons:
- Typo in line 331, ELBO, not EBLO.
- Eq (8) is not the same as in Algorithm 1. Shouldn't it be \z_u^{(l)} on the right side?
- I don't get why all datasets have a fix \tau = 0.25. Shouldn't it based on the feature size/graph structures?
- From my personal perspective, I would like to see the learned latent structure in main text, and have a comparison to the original graphs.

---

> ### Author Response · Authors · 2022-08-02
> **Response to Reviewer N5pw**
>
> Thank you for the positive comments and favorable assessment.
>
> **Q1: Why not use the approximation (used for all-pair) for message passing over existing structures?**
>
> In fact, as far as we can tell, the proposed approximation scheme could not be easily adapted to approximate the attention over existing structures (e.g., as in GAT) since it requires that the attention scores to be a regular matrix (i.e, the key/value pairs are shared by all queries). In contrast, the attentions from existing graph structures induce different key/value pairs for each query (since nodes have different neighbors), in which case one cannot leverage the shared summation terms for reducing the complexity as is done by Eqns 5 and 7. Therefore, we only consider approximation for the all-pair computation which can provide better expressiveness yet would otherwise be more computationally expensive.
>
> Of course our approach can still exploit existing input graph structures (when available) through the use of a relational bias (Eqn. 8) and an edge-level regularization loss (Eqn. 10) as complementary to the all-pair message passing on latent graphs.  This is much different than common GNNs that use the fixed structures for directly propagating node features.
>
> **Q2: Why is KNN still needed for graph-enhanced applications?**
>
> The GNN models used as baselines here still need input graph structures for layer-wise feature propagation, so we use a KNN approach to construct such input graphs, enabling them to work on these non-graph-based datasets. In a similar fashion, we also consider ALIGNS using the KNN graphs as a relational bias and edge-level regularization for fair comparison with GNN competitors, though ALIGNS could in principle work in situations without any input observed graphs. To verify this, we conducted extra experiments where we free ALIGNS from the input structures (i.e., not using any relational bias and edge loss) and we found the performance is still decent. Concretely, we have test accuracy 86.04 on mini-ImageNet and 64.23 on 20News-group, which is still competitive among all the models and suggests that ALIGNS can work smoothly even in non-graph settings.
>
> **Q3: Is there no dimension change during feature transformation?**
>
> We set the input and output dimensions of the feature transformation to be the same for simplicity. In Eqn. 8, $z$ is given by the computation by Eqn. 7, so $z$ should have the same dimension as $v$ even if we assume a dimension change for the feature transformation.
>
> **Q4: Why all the datasets have a fixed $\tau$?**
>
> Actually, we empirically found that a certain range of $\tau$'s values (small but not very small, e.g., $0.1\sim 1.0$) would work robustly across all the datasets. So, in practice, we set $\tau=0.25$ for simplicity.  In fact, this insensitivity of $\tau$ can be viewed as a strength since it need not be tuned carefully.
>
>
> For other minor issues, thanks for pointing out the typos and we have modified them in the new uploaded version. We will move the visualization from the supplementary to the main text once we have more space in the final version. Good suggestion.

---

### Official Review · Reviewer_CYMK · 2022-07-12

**Rating:** 7
**Confidence:** 3
**Soundness:** 4 excellent
**Presentation:** 4 excellent
**Contribution:** 3 good

**Summary:**

This paper introduces an all-pair message passing scheme based on a kernelized Gumbel-Softmax operator to reduce the complexity of message passing from quadratic (in number of nodes N) to linear time at the cost of some approximation error. The authors provide an analysis of the incurred error and present empirical comparisons with state-of-the-art methods.

**Questions:**

Please see Weaknesses section above.


### Post Rebuttal
Thank you for your response. This addresses my concern and my decision remains the same (Accept).

**Limitations:**

Yes (in the Appendix)

**Strengths And Weaknesses:**

## Strengths
* The paper is very well-written and organized
* The problem of efficient graph learning is highly relevant to the ML community
* Sound theoretical results are provided that bound the approximation error of the proposed scheme (Sec. 3.2)
* The presented method combines existing techniques (random features, reparameterization trick, Gumbel distribution sampling) in a novel way
* Empirical results and comparisons (Sec. 4) on common graph datasets support the improved effectiveness of the proposed approach
* Visualizations and sensitivity analysis of relevant variables (\tau) are provided in the appendix

## Weaknesses
* The method requires a parameter \tau to be tuned in advance. Is there any way to learn or automatically set \tau based on the theoretical analysis (e.g., mathematically balance the tradeoff mentioned in Line 178) so that ablation studies are not needed on an application specific basis?

---

> ### Author Response · Authors · 2022-08-02
> **Response to Reviewer CYMK**
>
> Thank you for the positive feedbacks. For the hyper-parameter $\tau$, it controls the smoothness of the learned structures. We found a range of its values (small but not very small, e.g., 0.1~1.0) works stably, while too small or too large can lead to performance degradation. Such a phenomenon is shared across all cases, so not much effort is needed for tuning $\tau$. In practice, we set it as 0.25 across all the experiments, which we found produced decent performance. Of course alternatively, we could consider it as a learnable model parameter and allow backpropagation to automatically update it with other model parameters. We leave this option for future work. Good suggestion.

---

### Author Response · Authors · 2022-08-02
**General Response**

Dear Area Chairs and Reviewers,

We appreciate the valuable feedback and constructive suggestions from the reviewers. Overall, the reviewers deem our paper well written, our method "novel" (CYMK) and "impressive" (N5pw, bJ9T), our theoretical analysis "sound" (CYMK) and "comprehensive" (Xhvj), our evalution results "solid" (CYMK) and "promising" (CYMK, N5pw). They also asserted that "the problem is highly relevant to the ML community" (CYMK) and the "method has potential for use in much wider range" (N5pw).

In the following individual response, we address all the raised questions and add some new experiment results to further strenghthen our contributions.

---

### Meta-Review · Area_Chair_2kWB · 2022-08-27

**Recommendation:** Accept
**Confidence:** Certain

**Metareview:**

The paper presents an approximate way to perform all-pair message passing within the context of GNNs. The paper's main contribution is a series of extensive empirical results and at the same time theoretical justification for the approach. All the reviewers liked the paper and noted the impressive scalability of the approach. Some technical questions/concerns were also addressed post rebuttal. This paper is recommended for acceptance.

**Award:**

No

---

### Decision · Program_Chairs · 2022-09-14

Accept